# DOMAIN GENERALIZATION VIA INDEPENDENT REGULARIZATION FROM EARLY-BRANCHING NETWORKS

## ABSTRACT

Learning domain-invariant feature representations is critical for achieving domain generalization, where a model is required to perform well on unseen domains. The critical challenge is that standard training often results in entangled domain-invariant and domain-specific features (see Figure 2). To address this issue, we use a dual-branching network to learn two features, one for the domain classification problem and the other for the original target classification problem, and the feature of the latter is required to be independent of the former. While this idea seems straightforward, we show that several factors need to be carefully considered for it to work effectively. In particular, we investigate different branching structures and discover that the common practice of using a shared base feature extractor with two lightweight prediction heads is detrimental to the performance. Instead, a simple early-branching architecture, where the domain classification and target classification branches share the first few blocks while diverging thereafter, leads to better results. Moreover, we also incorporate a random style augmentation scheme as an extension to further unleash the power of the proposed method, which can be seamlessly integrated into the dual-branching network by our loss terms. Such an extension gives rise to an effective domain generalization method. Experimental results show that the proposed method outperforms state-of-the-art domain generalization methods on various benchmark datasets.

## 1 INTRODUCTION

Domain generalization (DG) asks learned models to perform well on unseen domains, which lies its key in learning domain-invariant representations that are robust to domain shift (Ben-David et al., 2006). Standard training often results in entangled domain-invariant and domain-specific features, which hinders the model from generalizing to new domains. Existing methods address this issue by introducing various forms of regularization, such as adopting alignment (Muandet et al., 2013; Ghifary et al., 2016; Li et al., 2018b; Hu et al., 2020), using domain-adversarial training (Ganin et al., 2016; Li et al., 2018b; Yang et al., 2021; Li et al., 2018c), or developing meta-learning methods (Li et al., 2018a; Balaji et al., 2018; Dou et al., 2019; Li et al., 2019). Despite the success of these arts, DG remains challenging and is far from being solved. For example, as a recent study (Gulrajani & Lopez-Paz, 2021) suggests, under a rigorous evaluation protocol, it turns out that the naive empirical risk minimization (ERM) method (Vapnik, 1999), which aggregates training data from all domains and trains them in an end-to-end manner without additional efforts, can perform competitively against more elaborate alternatives. This observation indicates that a more effective approach might be needed to disentangle the domain-invariant and domain-specific features for better DG.

In this paper, we adopt a simple method by leveraging a conventional dual-branching network with one branch predicting image classes (target prediction) and another predicting domain labels. Regarding the features from the target and domain branches as domain-invariant and domain-specific representations, respectively, entanglement will result in an undesired situation where the domain-specific information is also encoded in the target branch, which will inevitably corrupt the prediction when the domain varies during inference. Thus, to explicitly disentangle the domain-invariant and domain-specific features, we impose a regularization to require the former to be independent of the latter. **This idea seems straightforward, but we show that several factors need to be carefully considered for it to work effectively**.

Particularly, we first investigate the structure of the dual-branching network and somehow surprisingly discover that the common practice of using a shared base feature extractor with two lightweight prediction heads (Chen et al., 2021; Atzmon et al., 2020) is detrimental to the performance. Instead, a simple early-branching architecture, where the domain classification branch and target classification branch share the first few blocks while diverging thereafter, yields the optimal results. Incorporating this discovery, we propose the basic form of the proposed method. Specifically, we employ Hilbert-Schmidt Information Criterion (HSIC) (Gretton et al., 2005; 2007) as a measurement of the feature independence [1] and use two sub-networks (branches) with only a few shared convolution blocks for target prediction task and domain prediction task. A glimpse of the basic form is shown in Figure 1.

Next, to further unleash the power of the proposed method, we suggest using domain augmentation to encourage the domain-invariant features to explore sufficient diversity of domain-specific representations. Precisely, we propose a new random style sampling (RDS) scheme that explores augmenting the domain types by incorporating features with randomly modified style statistics. In contrast to previous methods (Zhou et al., 2021; Li et al., 2022) that use mixing or adding noise to synthesize new domains, RDS can directly perturb the mean and variance of feature maps with a controllable perturbing strength. To seamlessly integrate the basic form and the augmentation strategy, we further propose subsequential loss terms to encourage the target branch to be invariant to the original and augmented representations and vice versa for the domain-specific branch.

Through our experimental studies, we illustrate, (1) the effectiveness of enforcing independence of the class and domain features within the early-branching design; (2) the advantages of the proposed RDS methods compared to existing solutions (Zhou et al., 2021; Li et al., 2022), and effectiveness of the adopted loss functions; (3) our complete method performs favorably against other state-of-the-art algorithms when evaluated in the current benchmark (Gulrajani & Lopez-Paz, 2021).

## 2 RELATED WORKS

Various methods have been proposed in the DG literature recently (Li et al., 2017; Motiian et al., 2017; Li et al., 2018b;a; Gong et al., 2019; Zhou et al., 2020a; Zhao et al., 2020; Li et al., 2019; Honarvar Nazari & Kovashka, 2020; Li et al., 2021; Zhou et al., 2021; Xu et al., 2021a; Kim et al., 2021; Wang et al., 2020; Bui et al., 2021; Yang et al., 2021; Li et al., 2022; Chen et al., 2022). Despite the varying details, current DG methods can be roughly categorized into a few categories by motivating intuition: invariant representation learning (Ganin et al., 2016; Li et al., 2017; 2018b;c; Shi et al., 2021), augmentation (Zhou et al., 2021; Li et al., 2022; Xu et al., 2021a; Li et al., 2021), and general machine learning algorithms such as meta-learning (Li et al., 2018a; Balaji et al., 2018; Dou et al., 2019; Li et al., 2019) and self-supervised learning (Carlucci et al., 2019a; Jeon et al., 2021; Kim et al., 2021). This section briefly reviews methods from the most relevant categories.

**Invariant representation learning.** The pioneer work (Ben-David et al., 2006) theoretically proved that if the features remain invariant across different domains, then they are general and transferable to different domains. Inspired by this theory, many recent arts aim to use deep networks to explore domain-invariant features. For example, (Ganin et al., 2016) train a domain-adversarial neural network (DANN) to obtain domain-invariant features by maximizing the domain classification loss. This idea is further explored by (Li et al., 2018b). They employ a maximum mean discrepancy constraint for the representation learning of an auto-encoder via adversarial training. Instead of directly obtaining the domain-invariant features, some arts (Khosla et al., 2012; Li et al., 2017) suggest decomposing the model parameters into domain-invariant and domain-specific parts and only using the domain-invariant parameters for prediction when confronting unseen domains. Recently, the task has been further explored at a gradient level. Koyama and Yamaguchi (Koyama & Yamaguchi, 2020) learn domain-invariant features by minimizing the variances of inter-domain gradients. Inspired by the fact that optimization directions should be similar across domains, (Shi et al., 2021) suggest maximizing the gradient inner products between domains to maintain the invariance. Different from previous approaches, our method is based on a simple intuition that domain and class features should be totally independent. By enforcing a straightforward independent constraint, our method can achieve comparable or even better performance against these arts.

---

[1]It is noteworthy that the early-branching structure can work with many independence measurements, and we use HSIC because it yields better performance. Please see Sec. 4.3 and B for details.

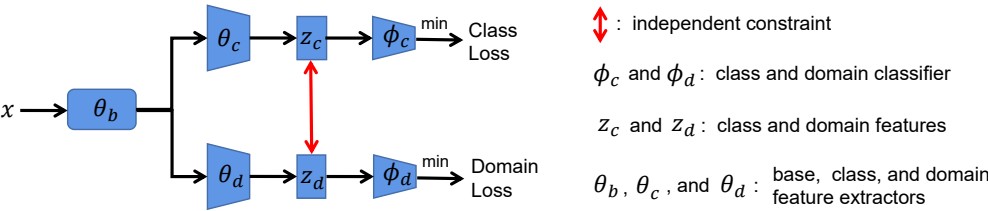

Figure 1: A glimpse of the proposed framework. The naive ERM is the framework without the independent constraint and the base feature extractor (i.e. $\theta_b = \mathbf{I}$, where $\mathbf{I}$ is the identity matrix).

**Augmentation.** Augmentation is one of the most popular strategies to improve the robustness of a model to novel domains. Besides the typical augmentation strategies, such as rotation, flipping, and cropping, there are some methods that are specially designed for the DG task. For example, (Yan et al., 2020) apply the mixup idea (Zhang et al., 2018) for DG by directly mixing images and labels from a batch. Instead of mixing in the image level, (Kim et al., 2021) propose to mixup the feature representations, (Xu et al., 2021a) suggest linearly interpolating the amplitude spectrums of different samples. How to new samples for training is also studied in the literature: generative modeling is used in (Zhou et al., 2020b) and (Carlucci et al., 2019b) to create new domain and domain-agnostic images. Inspired by the success in the related area, both works in (Zhou et al., 2021) and (Nam et al., 2021) use AdaIN (Huang & Belongie, 2017) to synthesize new domain information by mixing the statistics of the features. To further explore the idea, (Li et al., 2022) propose to add perturbations to the statistics based on their variances, and (Kang et al., 2022) suggest storing seen statistics in a queue and consistently generating distinct statistics for mixing. However, despite certain improvements, their augmented domain information may still suffer from homogeneity, thus preventing the model from further generalization. To obtain unlimited and diverse domain information, we assume the statistics of the features to follow normal distributions. Consequently, we can sample unlimited diverse domain information that derives from the original statistics. By integrating the augmented features in an effective framework, our method performs favorably against existing alternatives in the DomainBed benchmark (Gulrajani & Lopez-Paz, 2021).

## 3 OUR METHOD

**Problem setup.** Let $\mathcal{X}$ denote the image space, and $\mathcal{Y}$ represents the label space [2]. In the domain generalization (DG) problem, we are given $M$ source domains $\mathcal{D}_s = \{\mathcal{D}_1, \mathcal{D}_2, \cdots, \mathcal{D}_M\}$ that sampled from different probability distributions on the joint space $\mathcal{X} \times \mathcal{Y}$, where each $\mathcal{D}_m \in \mathcal{D}_s$ can be represented as $\mathcal{D}_m = \{(x_i^m, y_i^m)\}_{i=1}^{N_m}$ and $N_m$ is the number of data label pairs in the $m^{th}$ domain, the task is to learn a model from $\mathcal{D}_s$ for making predictions on an unseen $\mathcal{D}_{M+1}$ domain.

### 3.1 LEARNING DOMAIN-INVARIANT FEATURES VIA INDEPENDENCE REGULARIZATION

Our method is based on a simple principle: domain-invariant features should be independent of domain-specific features. This principle is based on the assumption that domain types and image classes are defined by different sets of visual cues and they can be estimated from different representations. The independent constraint will enforce the domain-invariant features to avoid focusing on the common cues, so that it can be less affected when the domain type varies during inference.

We implement this idea by learning two branches, one for domain classification and one for target classification, e.g., image classification. Formally, we follow the convention to use a shared base feature extractor, denoted as $\theta_b$. The two branches are implemented as cascades of feature extractors and classifiers, denoted as $\{\theta_c, \phi_c\}$ for the target branch and $\{\theta_d, \phi_d\}$ for the domain branch, respectively. Then we use Hilbert-Schmidt Information Criterion (HSIC) (Gretton et al., 2005; 2007) as the independence measure to regularize the statistic dependency between features from the domain and target extractors. We show in Sec. 4.3 that the adopted HSIC is more effective than other independent constraints such as orthogonal and correlation minimization. The learning objective (besides the common classification task) can be written as,

$$\min \text{HSIC}(z_c, z_d), \quad \text{s.t.} \quad z_c = \theta_c(\theta_b(x)), \quad z_d = \theta_d(\theta_b(x)) \tag{1}$$

---

[2]We only investigate DG on the classification task, thus instances in $\mathcal{Y}$ are the one-hot class labels.

Table 1: Evaluations of ERM and the proposed framework with different settings of $\theta_b$ in the unseen domain from (Li et al., 2017). Here $bck_{1,2,3,4}$ is the four blocks in a standard ResNet implementation with $bck_4$ close to the classifier. The common practice of using a shared base feature extractor with two lightweight prediction heads is detrimental to the performance.

| | art | cartoon | photo | sketch | avg |
|---|---|---|---|---|---|
| ERM | $78.0 \pm 1.3$ | $73.4 \pm 0.8$ | $94.1 \pm 0.4$ | $73.6 \pm 2.2$ | $79.8 \pm 0.4$ |
| $\theta_b = \mathbf{I}$ | $79.1 \pm 0.7$ | $74.1 \pm 1.1$ | $94.6 \pm 0.4$ | $74.8 \pm 0.6$ | $80.7 \pm 0.5$ |
| $\theta_b = bck_1$ | $79.3 \pm 0.9$ | $74.1 \pm 2.0$ | $94.8 \pm 0.5$ | $75.8 \pm 1.6$ | $81.0 \pm 0.6$ |
| $\theta_b = bck_{1,2}$ | $79.1 \pm 1.2$ | $73.2 \pm 1.8$ | $94.8 \pm 0.6$ | $74.9 \pm 1.3$ | $80.5 \pm 0.8$ |
| $\theta_b = bck_{1,2,3}$ | $78.3 \pm 0.2$ | $72.8 \pm 0.6$ | $94.9 \pm 0.9$ | $74.1 \pm 0.9$ | $80.0 \pm 0.2$ |
| $\theta_b = bck_{1,2,3,4}$ | $75.3 \pm 0.7$ | $72.1 \pm 0.2$ | $94.3 \pm 0.2$ | $71.5 \pm 0.6$ | $78.3 \pm 0.3$ |

where $\mathrm{HSIC}(z_c, z_d)$ measures the dependence between the representations $z_c$ and $z_d$. Note we do not use the domain classification branch when deploying. Thus the objective is only imposed on $\theta_c$ and $\theta_b$. Details regarding Eq. (1) can be found in the appendix.

**Observations: the branching location matters.** At first glance, the aforementioned algorithm seems to be quite straightforward, but as will be shown in the following parts, there are several factors that need to be carefully considered for it to work effectively. In this section, we focus on the first problem of how to choose the neural network architecture for the shared base feature extractor and two branches. The common practice is to use a deep sub-network as the shared base feature extractor and two lightweight branches with much fewer model parameters (Chen et al., 2021; Atzmon et al., 2020). Surprisingly, we empirically find that this strategy fails to produce a good performance. To systematically investigate this issue, we conduct an experiment by testing the performance with different branching locations.

We conduct experiments with ResNet18 (He et al., 2016) that has four blocks. We try to alter the model architecture by choosing block $k, k \in \{1, 2, 3, 4\}$ as the shared part. As for the corresponding two branches, we use $4 - k$ blocks plus an adaptive pooling layer and a fully-connected layer for feature generation (i.e. denoted as $\theta_c$ or $\theta_d$) and a classifier ($\phi_c$ or $\phi_d$) for each branch. We also consider a special case when no shared base feature extractor is used, which is equivalent to using two networks for domain and target prediction. We denote those cases as $\theta_b = \{\mathbf{I}, bck_1, bck_{1,2}, bck_{1,2,3}, bck_{1,2,3,4}\}$ and train the respective architecture with the loss in Eq. 1. Comparisons are shown in Table 1. We note that when the shared backbone is deep, i.e., $\theta_b = bck_{1,2,3,4}$, the performances are worse than the ERM baseline (79.8%). Meanwhile, using an earlier branching structure, e.g., $\theta_b = bck_1$ leads to better performances. Similar performances can also be observed when using different independent constraints (i.e. orthogonal and correlation minimization regularizations) and different network structures (i.e. smaller, larger convolution networks and vision transformers). Please refer to Sec. B in our appendix for more details.

This phenomenon can be understood from the role of $\theta_b$. Ideally, we hope the features for the target branch do not encode any domain information. This is expected to be achieved via the independence regularization in Eq. 1. However, if the domain and target branches share a significant part of the network, i.e., the base feature extractor $\theta_b$. Then $\theta_b$ will inevitably need to keep both domain and target information, making it harder to disentangle them at the respective branches. By contrast, if an early-branching architecture is used, we could minimize the need for encoding domain information in the pathway of target prediction, i.e., base feature extractor plus target prediction branch. Hereafter, we use $\theta_b = bck_1$ in the following discussion unless otherwise stated.

**t-SNE visualizations.** To further demonstrate the effectiveness of the aforementioned basic form, we present t-SNE visualizations of features extracted from the target feature extractor. We compare features obtained from our method and ERM. As plotted in Figure 2 (a) and (b), although both methods can separate different classes, our features tend to include less domain information as feature points from different domains are mixed. In comparison, we can see that features from different domains are clearly separated in the ERM case, indicating that ERM encodes more domain-specific information in their target features. These results validate that our regularization term can indeed help disentangle the domain-invariant and domain-specific features, and the proposed framework is more effective than the ERM method. Please see the appendix for more analysis.

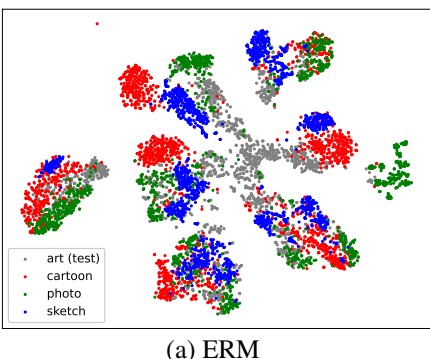 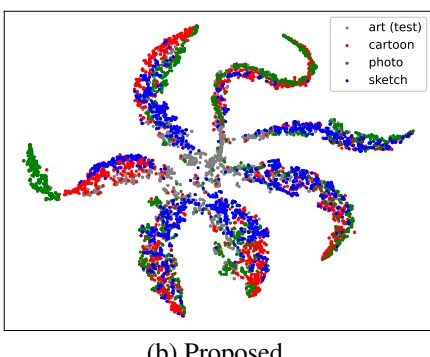

(a) ERM                                                 (b) Proposed

Figure 2: 2D t-SNE visualizations of target representations from the ERM model and our basic form. The PACS dataset (Li et al., 2017) is used with art as the unseen target domain. The seven clusters in (a) and (b) denote the corresponding classes. The domain information is more obvious in the target features from the ERM model, indicating that ERM tends to learn entangled domain-specific and domain-invariant features. In comparison, using the independent constraint can better disentangle the two features, resulting in less domain information in the target features.

## 3.2 INCORPORATING AUGMENTED DOMAINS

Although the independence regularization and a proper architecture can lead to certain improvements, we are aware that the limited number of domains in the training set could easily overfit the target features, thus hindering our method from achieving its full potential. To overcome this limitation, we proposed to use domain augmentation to further strengthen the independence between the features from the domain prediction branch and the target prediction branch. As suggested by a recent study (Zhou et al., 2021), random changes in the style statistics of feature maps, e.g., by mixing the mean and variance of feature maps from two samples, can be used to augment the limited image domains in the training set, and thus improve the generalization ability of the model substantially. In this section, we discuss an extension of the aforementioned basic form by further incorporating this kind of domain augmentation technique. In the following parts, we start by describing how our framework can be extended with domain augmentation in general, and then we discuss a new variant of domain augmentation scheme.

Formally, we use $\mathcal{A}(\cdot)$ to denote a domain augmentation operator that could modify image features in a way that changes the image style while preserving the target class information. In our design, $\mathcal{A}(\cdot)$ is applied to the features extracted from $\theta_b$ so that both the two branches will be affected. We use $f = \theta_b(x)$ and $\mathcal{A}(f)$ to denote the original image features and the domain-augmented image feature, respectively. Then the domain augmentation can be seamlessly incorporated into the proposed framework by the following loss functions.

The first is the classification loss $\mathcal{L}_{cls}$ that includes loss terms for both domain and target classification tasks. We use the cross entropy (CE) loss for the tasks, and $\mathcal{L}_{cls}$ can be represented as follows:

$$\mathcal{L}_{cls} = \mathrm{CE}(\phi_c(\theta_c(f)), y) + \mathrm{CE}(\phi_c(\theta_c(\mathcal{A}(f))), y) + \mathrm{CE}(\phi_d(\theta_d(f)), D_x), \tag{2}$$

where $D_x$ is the domain label of $x$. Note that in this setting, we choose not to treat the augmented domain equally as the true domain and thus do not assign domain labels for the augmented domain.

The second term is the sensitivity loss $\mathcal{L}_{sen}$ that regularizes the change of features from the target and domain branches under the domain augmentation. Intuitively, we expect the domain augmentation to have minimal impact on the target feature since it is supposed to be domain-invariant. On the other hand, we expect a sufficient change of features for the features for domain classification. Building upon this idea, we propose the following loss:

$$\mathcal{L}_{sen} = \|\theta_c(f) - \mathrm{MLP}(\theta_c(\mathcal{A}(f)))\|_1 + \|\mathrm{MLP}(\theta_c(f)) - \theta_c(\mathcal{A}(f))\|_1 -$$
$$([\|\theta_d(f) - \mathrm{MLP}(\theta_d(\mathcal{A}(f)))\|_1 - n]_- + [\|\mathrm{MLP}(\theta_d(f)) - \theta_d(\mathcal{A}(f))\|_1 - n]_-), \tag{3}$$

where $\|\cdot\|_1$ is the $L_1$ norm. As suggested by (Grill et al., 2020; Kim et al., 2021), we use $\|\mathrm{MLP}(\mathbf{a}) - \mathbf{b}\|$ to measure the compatibility between $\mathbf{a}$ and $\mathbf{b}$. It has shown that this scheme can prevent representation collapse (Grill et al., 2020; Kim et al., 2021). $[\cdot]_- = \min(\cdot, 0)$, and the margin $n$ is

considered a hyper-parameter which is set to be the maximum value of the $L_1$ distances computed from the current batch.

The last loss term $\mathcal{L}_{indp}$ is brought by the independent constraint. The independence between features from the two branches should be maintained under the augmentation. Therefore, we perform HSIC minimization regardless of whether original data or augmented data is used. Expanding the cases, this is equivalent to minimizing the following four-term loss:

$$
\begin{aligned}
\mathcal{L}_{indp} =& \text{HSIC}(\theta_c(f), \theta_d(f)) + \text{HSIC}(\theta_c(f), \theta_d(\mathcal{A}(f))) + \\
& \text{HSIC}(\theta_c(\mathcal{A}(f)), \theta_d(f)) + \text{HSIC}(\theta_c(\mathcal{A}(f)), \theta_d(\mathcal{A}(f))).
\end{aligned}
\tag{4}
$$

In summary, the overall loss $\mathcal{L}_{all}$ for the proposed algorithm can be expressed as,

$$
\mathcal{L}_{all} = \mathcal{L}_{cls} + \alpha \mathcal{L}_{sen} + \beta \mathcal{L}_{indp},
\tag{5}
$$

where $\alpha$ and $\beta$ are weight parameters.

## 3.3 A New Domain Augmentation Strategy

To better unleash the power of the proposed method, we design a new domain augmentation strategy dubbed random domain sampling (RDS). Our method is inspired by (Zhou et al., 2021), which shows that new domains can be obtained by mixing the style statistics of two different samples. In (Zhou et al., 2021), the style statistics are the mean $\mu \in \mathbb{R}^{B \times C}$ and standard derivation $\sigma^2 \in \mathbb{R}^{B \times C}$ of activations across all spatial locations and all samples in a batch, given a set of $B \times C \times H \times W$ sized feature maps (where $B, C, H,$ and $W$ are batch, channel, height, and width of the feature map).

However, modifying $\mu$ and $\sigma^2$ by directly mixing with other image style statistics may create more homogeneous features when confronting similar domain types within a batch. RDS solves this problem by perturbing the statistics with a controllable strength. Specifically, we first calculate the mean and variance of activations from each image, denoted as $\mu_i$ and $\sigma_i^2$. Then, we build a probabilistic model for $\mu_i$ and $\sigma_i^2$, which is done by assuming they follow the Gaussian distribution and then estimating the model parameters within a batch. Finally, we can sample new $\mu$ and $\sigma^2$ from the Gaussian distribution as new style statistics. Take $\mu$ as an example. The above procedure can be written as:

$$
\mu \sim \mathcal{N}(\mathbb{E}_\mu, \Sigma_\mu), \quad \text{s.t.} \quad \mathbb{E}_\mu = \frac{1}{B}\sum_{i=1}^{B}\mu_i, \quad \Sigma_\mu = \frac{1}{B}\sum_{i}^{B}(\mu_i - \mathbb{E}_\mu)(\mu_i - \mathbb{E}_\mu)^{\mathrm{T}},
\tag{6}
$$

To obtain inhomogeneous styles, we only accept sampled $\mu$ whose density is less than $\epsilon$. The accepted samples, denoted as $\hat{\mu}$ should satisfy

$$
\frac{1}{(2\pi)^{\frac{B \times C}{2}}|\Sigma_\mu|^{\frac{1}{2}}} \exp\left(-\frac{1}{2}(\hat{\mu} - \mathbb{E}_\mu)^{\mathrm{T}}\Sigma_\mu^{-1}(\hat{\mu} - \mathbb{E}_\mu)\right) < \epsilon,
\tag{7}
$$

where $\epsilon$ is sufficiently small (i.e. set to be 0.0001 in our implementation) so that the sampled $\hat{\mu}$ can be distinct from the original $\mu$. An illustration of the sampling strategy is shown in Figure 3. The same sampling process is applied to obtain $\hat{\sigma}$.

After sampling new style statistics, we follow the protocol in AdaIN (Huang & Belongie, 2017) by replacing the style statistics from the original representation with the sampled ones, and the final formulation for our domain augmentation can be written as,

$$
\mathcal{A}(f) = \hat{\sigma}\frac{f - \mu}{\sigma} + \hat{\mu},
\tag{8}
$$

where $f$ is the $C$ dimensional activation vectors at each spatial grid of a feature map. Different from the previous work (Li et al., 2022; Zhou et al., 2021) that specifically adds normal perturbations to their style statistics or borrows style statistics from other samples, RDS manages to

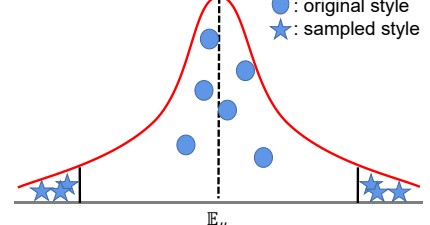

Figure 3: A simple demonstration of the RDS sampling strategy.

control how the sampled statistics deviated from the existing style statistics thanks to its probabilistic modeling of $\mu_i$ and $\sigma_i^2$. This prevents the augmentation strategy from creating more homogeneous domains and alleviates the drawback when confronting similar domain information in a batch.

Table 2: Evaluations on DomainBed (Gulrajani & Lopez-Paz, 2021). All methods are examined for 60 trials in each unseen domain. ERM + HSIC is our basic form introduced in Sec. 3.1 without the augmentation and with $\theta_b = blk_1$. Top5 accumulates the number of datasets where a method achieves the top 5 performances. Score here accumulates the numbers of the dataset where a specific art performs better than ERM. Best and second best results are presented in **bold** and undelined types.

| | PACS | VLCS | OfficeHome | TerraInc | DomainNet | Avg. | Top5 | Score |
|---|---|---|---|---|---|---|---|---|
| MMD | $81.3 \pm 0.8$ | $74.9 \pm 0.5$ | $59.9 \pm 0.4$ | $42.0 \pm 1.0$ | $7.9 \pm 6.2$ | 53.2 | 1 | 2 |
| RSC | $80.5 \pm 0.2$ | $75.4 \pm 0.3$ | $58.4 \pm 0.6$ | $39.4 \pm 1.3$ | $27.9 \pm 2.0$ | 56.3 | 0 | 2 |
| IRM | $80.9 \pm 0.5$ | $75.1 \pm 0.1$ | $58.0 \pm 0.1$ | $38.4 \pm 0.9$ | $30.4 \pm 1.0$ | 56.6 | 0 | 1 |
| ARM | $80.6 \pm 0.5$ | $75.9 \pm 0.3$ | $59.6 \pm 0.3$ | $37.4 \pm 1.9$ | $29.9 \pm 0.1$ | 56.7 | 0 | 2 |
| DANN | $79.2 \pm 0.3$ | $76.3 \pm 0.2$ | $59.5 \pm 0.5$ | $37.9 \pm 0.9$ | $31.5 \pm 0.1$ | 56.9 | 1 | 1 |
| GroupGRO | $80.7 \pm 0.4$ | $75.4 \pm 1.0$ | $60.6 \pm 0.3$ | $41.5 \pm 2.0$ | $27.5 \pm 0.1$ | 57.1 | 0 | 2 |
| CDANN | $80.3 \pm 0.5$ | $76.0 \pm 0.5$ | $59.3 \pm 0.4$ | $38.6 \pm 2.3$ | $31.8 \pm 0.2$ | 57.2 | 0 | 2 |
| VREx | $80.2 \pm 0.5$ | $75.3 \pm 0.6$ | $59.5 \pm 0.1$ | $\mathbf{43.2 \pm 0.3}$ | $28.1 \pm 1.0$ | 57.3 | 1 | 2 |
| CAD | $81.9 \pm 0.3$ | $75.2 \pm 0.6$ | $60.5 \pm 0.3$ | $40.5 \pm 0.4$ | $31.0 \pm 0.8$ | 57.8 | 1 | 2 |
| CondCAD | $80.8 \pm 0.5$ | $76.1 \pm 0.3$ | $61.0 \pm 0.4$ | $39.7 \pm 0.4$ | $31.9 \pm 0.7$ | 57.9 | 0 | 4 |
| MTL | $80.1 \pm 0.8$ | $75.2 \pm 0.3$ | $59.9 \pm 0.5$ | $40.4 \pm 1.0$ | $35.0 \pm 0.0$ | 58.1 | 0 | 2 |
| ERM | $79.8 \pm 0.4$ | $75.8 \pm 0.2$ | $60.6 \pm 0.2$ | $38.8 \pm 1.0$ | $35.3 \pm 0.1$ | 58.1 | 0 | - |
| MixStyle | $\mathbf{82.6 \pm 0.4}$ | $75.2 \pm 0.7$ | $59.6 \pm 0.8$ | $40.9 \pm 1.1$ | $33.9 \pm 0.1$ | 58.4 | 1 | 2 |
| ERM + HSIC | $81.0 \pm 0.6$ | $76.1 \pm 0.1$ | $60.7 \pm 0.3$ | $38.8 \pm 1.1$ | $35.4 \pm 0.1$ | 58.4 | 1 | 4 |
| MLDG | $81.3 \pm 0.2$ | $75.2 \pm 0.3$ | $60.9 \pm 0.2$ | $40.1 \pm 0.9$ | $35.4 \pm 0.0$ | 58.6 | 1 | 4 |
| Mixup | $79.2 \pm 0.9$ | $76.2 \pm 0.3$ | $61.7 \pm 0.5$ | $42.1 \pm 0.7$ | $34.0 \pm 0.0$ | 58.6 | 2 | 3 |
| Fishr | $81.3 \pm 0.3$ | $76.2 \pm 0.3$ | $60.9 \pm 0.3$ | $\underline{42.6 \pm 1.0}$ | $34.2 \pm 0.3$ | 59.0 | 2 | 4 |
| SagNet | $81.7 \pm 0.6$ | $75.4 \pm 0.8$ | $\underline{62.5 \pm 0.3}$ | $40.6 \pm 1.5$ | $35.3 \pm 0.1$ | 59.1 | 1 | 3 |
| SelfReg | $81.8 \pm 0.3$ | $76.4 \pm 0.7$ | $62.4 \pm 0.1$ | $41.3 \pm 0.3$ | $34.7 \pm 0.2$ | 59.3 | 2 | 4 |
| Fish | $82.0 \pm 0.3$ | $\mathbf{76.9 \pm 0.2}$ | $62.0 \pm 0.6$ | $40.2 \pm 0.6$ | $35.5 \pm 0.0$ | 59.3 | 3 | 5 |
| CORAL | $81.7 \pm 0.0$ | $75.5 \pm 0.4$ | $62.4 \pm 0.4$ | $41.4 \pm 1.8$ | $\underline{36.1 \pm 0.2}$ | 59.4 | 2 | 4 |
| SD | $81.9 \pm 0.3$ | $75.5 \pm 0.4$ | $\mathbf{62.9 \pm 0.2}$ | $42.0 \pm 1.0$ | $\mathbf{36.3 \pm 0.2}$ | $\underline{59.7}$ | 4 | 4 |
| Ours | $\underline{82.4 \pm 0.4}$ | $76.5 \pm 0.4$ | $62.2 \pm 0.1$ | $\mathbf{43.2 \pm 1.3}$ | $34.9 \pm 0.1$ | $\mathbf{59.8}$ | 4 | 4 |

# 4 EXPERIMENTS

## 4.1 DATASETS AND DETAILS

**Datasets.** To evaluate the effectiveness of our method, we conduct extensive experiments on five DG benchmarks, namely PACS (Li et al., 2017), VLCS (Fang et al., 2013), OfficeHome (Venkateswara et al., 2017), TerraInc (Beery et al., 2018), and DomainNet (Peng et al., 2019). Specifically, PACS consists of 9,991 images which can be divided into 7 classes. This dataset is the most commonly used DG benchmark due to its large distributional shift across 4 domains including art painting, cartoon, photo, and sketch; VLCS collects a total of 10,729 images from 4 different datasets (i.e. PASCAL VOC 2007 (Everingham et al., 2010), LabelMe (Russell et al., 2008), Caltech (Fei-Fei et al., 2004), and Sun (Xiao et al., 2010)) which can be categorized into 5 classes. Images from different datasets are taken under different views and considered different domains in DG; OfficeHome is an object recognition dataset that consists of images in office and home environments, and there are a total of 15,588 images from 65 classes in it. Images from OfficeHome can be divided into 4 domains including artistic, clipart, product, and real world; TerraInc contains 24,788 images of wild animals from 10 classes. Those images are taken from 4 different locations (i.e. L100, L38, L43, L46), and the locations are regarded as the varying domains in DG; DomainNet contains 586,575 images. There are a total of 345 classes included in DomainNet whose domains can be depicted in 6 styles (i.e. clipart, infograph, painting, quickdraw, real, and sketch).

**Implementation details.** We follow the prevalent design by using ResNet18 (ImageNet pretrained) as the backbone, which could enlarge the gaps in DG compared to larger models (Ye et al., 2022). The hyper-parameters in our model include $\alpha$ and $\beta$ in Eq. (5). We use a dynamic range of $\alpha \in [0.03, 0.3]$ and $\beta \in [0.1, ]$ for our experiments (analysis regarding these two hyper-parameters are given in the appendix). Settings for all the compared methods are set according to (Gulrajani & Lopez-Paz, 2021).

**Evaluation details.** For all the compared arts, we follow the default settings in (Gulrajani & Lopez-Paz, 2021). The leave-one-out strategy is used for evaluations. That is, for all the datasets, one domain is selected as the held-out for test and the remaining domains are treated as source domains for training. We evaluate all the methods 60 times in each source domain. The learning rates, augmentation strategies, random seeds, and batch sizes are all dynamically set for the compared

Table 3: Ablation studies regarding the effectivenesses of the proposed RDS augmentation strategy, the adopted losses, and the adopted HSIC independent constraint. Experiments are conducted in PACS (Li et al., 2017) with the leave-one-out training-test strategy. Here RDS$^-$ is our RDS without Eq. (7); $\mathcal{L}_{indp}^{oth}$ and $\mathcal{L}_{indp}^{corr}$ are variants of our method that use orthogonal and correlation as independent measurements.

| | model | art | cartoon | photo | sketch | avg |
|---|---|---|---|---|---|---|
| | Ours | $81.3 \pm 1.4$ | $75.7 \pm 0.2$ | $94.6 \pm 0.3$ | $78.1 \pm 0.7$ | $82.4 \pm 0.4$ |
| Augments | Ours w/o augmentation | $79.3 \pm 0.9$ | $74.1 \pm 2.0$ | $94.8 \pm 0.5$ | $75.8 \pm 1.6$ | $81.0 \pm 0.6$ |
| | Ours w/ MixStyle | $80.8 \pm 0.8$ | $75.1 \pm 0.3$ | $94.4 \pm 0.5$ | $77.4 \pm 1.5$ | $81.9 \pm 0.3$ |
| | Ours w/ DSU | $81.3 \pm 0.5$ | $75.1 \pm 0.8$ | $94.2 \pm 0.4$ | $76.3 \pm 0.5$ | $81.7 \pm 0.3$ |
| | Ours w/ RDS$^-$ | $80.7 \pm 0.7$ | $75.4 \pm 0.5$ | $94.6 \pm 0.9$ | $76.6 \pm 1.7$ | $81.9 \pm 0.7$ |
| Losses | Ours w/o $\mathcal{L}_{indp}$ & $\mathcal{L}_{sen}$ | $79.9 \pm 0.7$ | $75.6 \pm 0.4$ | $94.2 \pm 0.2$ | $76.2 \pm 0.8$ | $81.5 \pm 0.3$ |
| | Ours w/o $\mathcal{L}_{indp}$ | $80.5 \pm 0.5$ | $76.2 \pm 0.4$ | $94.0 \pm 0.2$ | $76.4 \pm 0.2$ | $81.8 \pm 0.1$ |
| | Ours w/o $\mathcal{L}_{sen}$ | $79.3 \pm 0.4$ | $75.4 \pm 0.9$ | $94.5 \pm 0.2$ | $78.3 \pm 1.7$ | $81.9 \pm 0.5$ |
| Constraints | Ours w/ $\mathcal{L}_{indp}^{oth}$ | $80.6 \pm 1.2$ | $75.4 \pm 0.5$ | $94.4 \pm 0.1$ | $77.9 \pm 1.4$ | $82.1 \pm 0.3$ |
| | Ours w/ $\mathcal{L}_{indp}^{corr}$ | $81.2 \pm 0.6$ | $75.8 \pm 0.2$ | $94.2 \pm 0.6$ | $77.7 \pm 0.5$ | $82.2 \pm 0.3$ |

methods in reasonable ranges, and the iteration step is fixed as 5,000 for each training. During training, we split the examples from training domains to 8:2 (train:val) and evaluate the models every 300 iteration steps. Note the training and validation samples are also dynamically selected among different training trials. During test, we select the model that performs the best in the validation samples and test it on the whole held-out domain, which is also the "training-domain validate set" model selection method in (Gulrajani & Lopez-Paz, 2021). For each domain in different datasets, the final performance is the average accuracy computed from the 60 trials, and the performance in this dataset is the average of the corresponding domains. To ensure fair comparisons, all the methods are evaluated in a same device (8 Nvidia Tesla v100 GPUs and each with 32G memory).

## 4.2 EXPERIMENTAL RESULTS

Evaluation results are shown in Table 2. We observe that the naive ERM method obtains comparable results against existing arts. In fact, as a strong baseline, ERM is ranked 11th place in the term of average accuracy, and only half of the state-of-the-arts can outperform ERM in most datasets (i.e. Score $\geq$ 3). Specially, we note that when using sophisticated designs to learn domain-invariant features on the basis of ERM, the average performances seem to decrease in most cases for IRM (Arjovsky et al., 2019), MMD (Li et al., 2018b), DANN (Ganin et al., 2016), CDANN (Li et al., 2018c), and RSC (Huang et al., 2020). By contrast, when ERM is imposed with a straightforward independent constraint (i.e. ERM + HSIC in the table with $\theta_b = blk_1$, which is detailed in Sec. 3.1), the average accuracy increases in most datasets, and the corresponding overall performance is also better than the naive ERM method. The results validate that the simple independent constraint is more effective at disentangling domain-invariant and domain-specific features than some pioneer arts.

Moreover, compared with alternatives that either use augmentations (i.e. Mixup (Yan et al., 2020), SagNet (Nam et al., 2021), SelfReg (Kim et al., 2021), MixStyle (Zhou et al., 2021), and CAD (Ruan et al., 2022)) or gradient-level constraints (i.e. , Fish (Shi et al., 2021), SD (Pezeshki et al., 2021), and Fishr (Rame et al., 2022)), our method consistently shows comparable or better performance in most datasets which obtains the top 5 performances in 4 out 5 datasets and is ranked 1st place in the term of average accuracy of all 5 benchmarks. The results validate the effectiveness of our method. We present results of average accuracy in each domain from different datasets in the appendix.

## 4.3 ABLATION STUDIES

All experiments in this section are conducted on the PACS (Li et al., 2017) benchmark with the same settings illustrated in Sec. 4.1. Please refer to the appendix for more ablation studies.

**Effectiveness of our RDS augmentation strategy.** Our augmentation strategy RDS aims to further strengthen the generalizability of our method by preventing the target features from overfitting to the limited training domains. We first compare our model against the baseline model that does not use any augmentation strategies (i.e. Ours w/o augmentation in Table 3). Note when the augmentation step is disabled, the overall algorithm reduces to the framework introduced in Sec. 3.1 with $\theta_b = bck_1$

where $\mathcal{L}_{sen}$, related parts in $\mathcal{L}_{cls}$, and $\mathcal{L}_{indp}$ in $\mathcal{L}_{all}$ are also disabled. As shown in the first row in Table 3, when enabling augmentation, our method can outperform the baseline model by a large margin in almost all domains except for "photo", which might be due to the ImageNet pretraining (Xu et al., 2021b). Note that the improvement for "sketch" is more pronounced than others, indicating our RDS can generalize to a target domain that significantly deviates from the source domains.

We also compare variants of our method by replacing RDS with other augmentations based on AdaIN (i.e. Ours w/ MixStyle (Zhou et al., 2021) and Ours w/ DSU (Li et al., 2022)). As listed in the 3rd and 4th rows, RDS consistently outperforms augmentations from (Zhou et al., 2021) and (Li et al., 2022) on average. We postulate that this is due to RDS's capability of synthesizing more diverse domains thanks to the sampling scheme. Moreover, to test if RDS can bring improvements against the naive sampling strategy, we evaluate another variant of our method by disabling the process depicted in Eq. (7) (i.e. Ours w/ RDS$^-$). Results in the 5th row show that naive sampling is not as effective as the proposed sampling strategy, which is not surprising since styles from the naive sampling may not be diverse enough compared to that from RDS. Note that the improvements are more noticeable than others when it comes to the "sketch" domain, which might be due to the reason that the rest three domains are close (Zhou et al., 2021), thus requiring less diverse information for generalizing. All these results validate the effectiveness of the proposed RDS against existing methods.

**Effectiveness of the losses.** In our full algorithm, there are two additional loss terms in addition to the main classification losses: the sensitivity loss $\mathcal{L}_{sen}$ and the independent constraint loss $\mathcal{L}_{indp}$. We study their effectiveness by disabling them separately in our framework. Results are listed in the 6th, 7th and 8th rows in Table 3. We observe that disabling either one of them can lead to performance decreases over our original design, demonstrating that each loss can independently contribute to the good performance of our method. Meanwhile, we also note that when the two loss terms are both disabled, the corresponding model is inferior to other variants, meaning these two terms are complementary in our framework and can further improve the generalization.

**Effectiveness of the adopted HSIC independent constraint.** We minimize the HSIC value between the target and domain features (i.e. $z_c$ and $z_d$) to reduce their dependency. As there exist other dependence measurements, such as orthogonal and correlation, one may wonder if HSIC is a better choice compared to them. To answer this question, we conduct ablation studies by replacing the HSIC measurement with the other two criteria. The orthogonal constraint requires the multiplication of the two features to be zero (i.e. Ours w/ $\mathcal{L}_{indp}^{oth}$); the correlation constraint minimizes the cross-correlation matrix computed between $z_c$ and $z_d$, and we use the implementation from (Zbontar et al., 2021) for this experiment (i.e. Ours w/ $\mathcal{L}_{indp}^{corr}$). Experimental results listed in the 9th and 10th rows in Table 3 suggest that the adopted HSIC independent constraint is more effective than the other two. We thus use the HSIC criterion in this work.

## 5 DISCUSSIONS AND CONCLUSION

**Limitations.** The major limitation of this work is that both the target and domain labels are required during training. Since we need the domain-specific features to partly guide the target representation learning, it is non-trivial to directly extend our method to occasions where domain labels are unavailable. We leave the exploration of such a setting to future work. Another drawback of the proposed method is that the framework requires another thread for the domain-specific classification task during training, bringing more parameters to the overall pipeline. Although we can use the same amount of parameters for the target classification task when deploying the model, training both two tasks will inevitably bring extra effort to the system. A promising improvement for the method is to use much fewer parameters to obtain comparable performance against the original implementation.

**Conclusion.** In this paper, we propose a method for domain generalization by imposing independent regularization for the features learned for domain classification and target prediction. We identify that several factors need to be considered to make this simple idea work effectively: first, an early branching network architecture is more effective than the commonly used deep backbone and lightweight predicting branch structure; second, incorporating augmented domains can further improve the learning outcome, and we introduce a carefully designed domain augmentation method to benefit the generalizability. These two findings are seamlessly integrated by the subsequential loss terms. Through extensive experiments on five benchmark datasets, we show that the proposed algorithm can obtain competitive performance against state-of-the-art alternatives.

## REPRODUCIBILITY STATEMENT

For data pre-processing steps, network configurations, and detailed learning settings, we use the existing benchmark (Gulrajani & Lopez-Paz, 2021) for all the implementations. Hyper-parameters specially used in this work, including $\alpha$ and $\beta$ used in Eq. 5, are fixed for all experiments and are analyzed in Sec. C.4. Please also refer to Table 12 for details. We will release the code upon acceptance.

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

## A    COMPUTING HSIC

The Hilbert-Schmidt Information Criterion (HSIC) (Gretton et al., 2005; 2007) is a non-parametric method for estimating the statistical dependence between samples of two variables. Given two batches of vectors (i.e. $z_c$ and $z_d$ from the target and domain branches), to compute the linear-kernel HSIC($z_c, z_d$), we first normalize the given vectors: $\bar{z}_c = \frac{z_c}{\|z_c\|}$, and $\bar{z}_d = \frac{z_d}{\|z_d\|}$, where $\|\cdot\|$ denotes the $L_2$ norm. Then, we can obtain two linear kernel matrices: $K = \bar{z}_c\bar{z}_c^T$, $L = \bar{z}_d\bar{z}_d^T$. Finally, the HSIC is calculated as the scaled Hilbert-Schmidt norm of their cross-covariance matrix:

$$\text{HSIC}(z_c, z_d) = \frac{1}{(B-1)^2}\mathbf{tr}(KHLH), \tag{9}$$

where $B$ is the batch size, $H = \mathbf{I} - \frac{1}{B}$ is a centering matrix, and $\mathbf{tr}(\cdot)$ computes the trace of the given matric. We refer the reader to the original paper (Gretton et al., 2005; 2007) for detailed derivations.

## B    MORE EVALUATIONS OF THE EARLY-BRANCHING STRUCTURE

In the manuscript, we suggest that despite the effectiveness of the adopted independent constraint, an early-branching network structure is essential to the overall performance. This phenomenon can be explained by the role of the base feature extractor $\theta_b$. If the two branches share a significant part of $\theta_b$. Then $\theta_b$ will be asked to provide both domain and target information, making it harder to disentangle them at the respective branches. However, if an early-branching architecture is used, we could minimize the need for encoding domain information in the pathway of target prediction.

### B.1    EARLY-BRANCHING STRUCTURE WITH DIFFERENT STRUCTURES.

To further verify our hypothesis, we conduct more evaluations with different network structures in this section. **(1)** We evaluate the independent constraint on a simple network that consists of 4 convolution layers. Here the domain of $\theta_b$ (i.e. $\theta_b = \{\mathbf{I}, bck_1, bck_{1,2}, bck_{1,2,3}, bck_{1,2,3,4}\}$) is similar to that in the manuscript with $bck_4$ closest to the classification head. We use the Rotated MNIST dataset (Ghifary et al., 2015) (i.e. RMNIST) in this experiment. RMNIST is originally designed for the handwritten digit recognition task which consists of 70,000 samples and can be categorized into 6 domains by its rotation degree: $\{0, 15, 30, 45, 60, 75\}$. **(2)** We use a larger ResNet50 model with more parameters to evaluate the adopted independent constraint. **(3)** We adopt vision transformers to evaluate the early-branching structure. Specifically, we use the DeiT (Touvron et al., 2021) and a T2T (Yuan et al., 2021) transformers that both with 12 attention layers in our evaluations. Considering the consistency with the aforementioned network structures, we divide the 12 attention layers into 4 blocks and each with 3 attention layers. In these two experiments, the PACS dataset (Li et al., 2017) is utilized. Note in all three evaluations, the experimental settings are the same as that in the benchmark (Gulrajani & Lopez-Paz, 2021).

Results are listed in Table 4, 5, and 6. We observe that we can obtain the optimal results on both the smaller (i.e. 4 convs), larger (i.e. ResNet50) convolution networks, and transformers (i.e. DeiT (Touvron et al., 2021) and T2T (Yuan et al., 2021)) with an early-branching structure, and the results decrease thereafter. These experiments further validate our hypothesis.

### B.2    EARLY-BRANCHING STRUCTURE WITH DIFFERENT INDEPENDENT CONSTRAINTS.

Except for the adopted HSIC, there also exist other dependency measurements, such as orthogonal and correlation. To verify if the early-branching structure can boost the performance despite the

Table 4: Evaluations of the early-branching structure with a lightweight four layer network in the unseen domain from RMNIST (Ghifary et al., 2015). Here $bck_{1,2,3,4}$ is the four convolution layers with $bck_4$ close to the classifier. The proposed framework achieves the optimal performances with an early-branching network structure (i.e. $\theta_b = bck_1$ or $\theta_b = \mathbf{I}$).

| | 0 | 15 | 30 | 45 | 60 | 75 | avg |
|---|---|---|---|---|---|---|---|
| ERM | $95.7 \pm 0.2$ | $98.6 \pm 0.1$ | $98.9 \pm 0.1$ | $99.0 \pm 0.1$ | $98.8 \pm 0.0$ | $96.0 \pm 0.1$ | $97.9 \pm 0.1$ |
| $\theta_b = \mathbf{I}$ | $95.8 \pm 0.0$ | $98.1 \pm 0.1$ | $98.8 \pm 0.0$ | $99.4 \pm 0.1$ | $99.0 \pm 0.0$ | $96.6 \pm 0.2$ | $98.0 \pm 0.1$ |
| $\theta_b = bck_1$ | $96.2 \pm 1.0$ | $98.4 \pm 0.1$ | $99.1 \pm 0.1$ | $98.9 \pm 0.1$ | $99.0 \pm 0.1$ | $96.3 \pm 0.5$ | $98.0 \pm 0.1$ |
| $\theta_b = bck_{1,2}$ | $91.4 \pm 1.0$ | $98.2 \pm 0.0$ | $98.6 \pm 0.1$ | $99.0 \pm 0.1$ | $98.8 \pm 0.1$ | $95.0 \pm 0.1$ | $96.8 \pm 0.1$ |
| $\theta_b = bck_{1,2,3}$ | $88.7 \pm 0.3$ | $97.5 \pm 0.1$ | $98.5 \pm 0.0$ | $98.9 \pm 0.0$ | $98.6 \pm 0.1$ | $94.3 \pm 0.1$ | $96.1 \pm 0.1$ |
| $\theta_b = bck_{1,2,3,4}$ | $90.4 \pm 0.6$ | $97.0 \pm 0.3$ | $98.4 \pm 0.0$ | $98.5 \pm 0.1$ | $98.6 \pm 0.1$ | $92.8 \pm 0.1$ | $95.9 \pm 0.2$ |

Table 5: Evaluations of the early-branching structure with a Resnet50 backbone in the unseen domain from (Li et al., 2017). Here $bck_{1,2,3,4}$ is the four layers in a standard ResNet50 implementation with $bck_4$ close to the classifier. The proposed framework achieves the optimal performance with an early-branching network structure (i.e. $\theta_b = bck_1$ or $\theta_b = \mathbf{I}$).

| | art | cartoon | photo | sketch | avg |
|---|---|---|---|---|---|
| ERM | $85.1 \pm 1.3$ | $78.0 \pm 1.2$ | $97.3 \pm 0.1$ | $72.2 \pm 1.6$ | $83.1 \pm 0.9$ |
| $\theta_b = \mathbf{I}$ | $85.8 \pm 0.9$ | $77.6 \pm 0.1$ | $97.2 \pm 0.2$ | $77.7 \pm 0.4$ | $84.5 \pm 0.3$ |
| $\theta_b = bck_1$ | $85.6 \pm 1.0$ | $77.9 \pm 1.7$ | $97.5 \pm 0.4$ | $77.7 \pm 0.1$ | $84.7 \pm 0.5$ |
| $\theta_b = bck_{1,2}$ | $84.2 \pm 0.7$ | $77.7 \pm 1.1$ | $96.3 \pm 0.2$ | $73.5 \pm 0.5$ | $82.9 \pm 0.4$ |
| $\theta_b = bck_{1,2,3}$ | $81.2 \pm 1.2$ | $77.1 \pm 0.6$ | $97.1 \pm 0.1$ | $72.6 \pm 0.7$ | $82.0 \pm 0.5$ |
| $\theta_b = bck_{1,2,3,4}$ | $85.4 \pm 1.6$ | $78.0 \pm 0.8$ | $97.1 \pm 0.2$ | $66.4 \pm 2.4$ | $81.7 \pm 0.7$ |

varying independent constraints, we conduct experiments by replacing HSIC with other criteria. All the experiments are conducted on the PACS dataset (Li et al., 2017) with the ResNet18 backbone. As shown in Table 7, the early-branching structure consistently outperforms the common practice of using lightweight prediction heads when the HSIC measurement is replaced with orthogonal and correlation criteria. These results demonstrate that dependency measurement is not a crucial factor in the early-branching structure.

## B.3 EARLY-BRANCHING STRUCTURE WITH AUGMENTATION.

Moreover, since our method incorporates a random domain augmentation scheme into the dual-branching network, one may wonder if the early-branching network structure is still the optimal choice in our method. To answer this question, we conduct experiments on the PACS dataset (Li et al., 2017) using both the ResNet18 and ResNet50 networks (He et al., 2016). Note in these experiments,

Table 6: Evaluations of the early-branching structure with vision transformer backbones (i.e. DeiT Touvron et al. (2021) and T2T (Yuan et al., 2021)) in the unseen domain from (Li et al., 2017). Here $bck_{1,2,3,4}$ is the 12 attention layers in the transformers, with each block corresponding to 3 attention layers and $bck_4$ close to the classifier. The proposed framework achieves the optimal performance with an early-branching network structure (i.e. $\theta_b = bck_1$ or $\theta_b = \mathbf{I}$).

| | | art | cartoon | photo | sketch | avg |
|---|---|---|---|---|---|---|
| | ERM-ViT | $87.3 \pm 0.4$ | $81.5 \pm 0.3$ | $98.7 \pm 0.4$ | $74.5 \pm 1.9$ | $85.5 \pm 0.6$ |
| | $\theta_b = \mathbf{I}$ | $87.7 \pm 0.6$ | $81.2 \pm 0.4$ | $98.3 \pm 0.2$ | $76.0 \pm 2.5$ | $85.8 \pm 0.4$ |
| Deit | $\theta_b = bck_1$ | $88.4 \pm 0.8$ | $80.5 \pm 1.4$ | $98.8 \pm 0.3$ | $77.8 \pm 0.9$ | $86.4 \pm 0.2$ |
| | $\theta_b = bck_{1,2}$ | $86.9 \pm 0.7$ | $80.6 \pm 0.9$ | $98.6 \pm 0.0$ | $74.6 \pm 1.2$ | $85.2 \pm 0.4$ |
| | $\theta_b = bck_{1,2,3}$ | $86.5 \pm 0.2$ | $78.0 \pm 0.8$ | $98.4 \pm 0.3$ | $77.3 \pm 0.3$ | $85.1 \pm 0.3$ |
| | $\theta_b = bck_{1,2,3,4}$ | $83.2 \pm 2.0$ | $77.8 \pm 1.1$ | $97.9 \pm 0.4$ | $75.5 \pm 0.5$ | $83.6 \pm 0.5$ |
| | ERM-ViT | $83.5 \pm 0.2$ | $75.8 \pm 0.7$ | $97.7 \pm 0.3$ | $76.9 \pm 0.8$ | $83.5 \pm 0.3$ |
| | $\theta_b = \mathbf{I}$ | $86.0 \pm 0.9$ | $76.0 \pm 2.2$ | $98.4 \pm 0.1$ | $75.7 \pm 1.8$ | $84.0 \pm 1.0$ |
| T2T | $\theta_b = bck_1$ | $85.1 \pm 0.9$ | $75.6 \pm 1.4$ | $98.4 \pm 0.1$ | $77.3 \pm 0.5$ | $84.1 \pm 0.5$ |
| | $\theta_b = bck_{1,2}$ | $82.8 \pm 0.9$ | $77.4 \pm 1.6$ | $97.9 \pm 0.5$ | $74.7 \pm 1.5$ | $83.2 \pm 0.3$ |
| | $\theta_b = bck_{1,2,3}$ | $81.6 \pm 0.4$ | $78.4 \pm 1.5$ | $97.5 \pm 0.6$ | $75.4 \pm 1.3$ | $83.2 \pm 0.5$ |
| | $\theta_b = bck_{1,2,3,4}$ | $82.6 \pm 0.6$ | $73.6 \pm 0.8$ | $98.0 \pm 0.2$ | $74.2 \pm 1.2$ | $82.1 \pm 0.3$ |

Table 7: Evaluations of early-branching structure with different independent constraints in the unseen domain from (Li et al., 2017) using the ResNet18 backbone. Here $bck_{1,2,3,4}$ is the four layers in a standard ResNet implementation with $bck_4$ close to the classifier.

| | | art | cartoon | photo | sketch | avg |
|---|---|---|---|---|---|---|
| | ERM | $8.0 \pm 1.3$ | $73.4 \pm 0.8$ | $94.1 \pm 0.4$ | $73.6 \pm 2.2$ | $79.8 \pm 0.4$ |
| Orthogonal | $\theta_b = \mathbf{I}$ | $80.6 \pm 0.6$ | $73.9 \pm 1.8$ | $94.2 \pm 0.1$ | $73.0 \pm 0.8$ | $80.4 \pm 0.4$ |
| | $\theta_b = bck_1$ | $78.9 \pm 1.3$ | $74.3 \pm 0.6$ | $94.3 \pm 0.5$ | $74.7 \pm 1.1$ | $80.5 \pm 0.3$ |
| | $\theta_b = bck_{1,2}$ | $79.5 \pm 0.9$ | $73.8 \pm 1.9$ | $94.5 \pm 0.4$ | $73.2 \pm 0.4$ | $80.2 \pm 0.4$ |
| | $\theta_b = bck_{1,2,3}$ | $75.2 \pm 1.4$ | $72.6 \pm 1.0$ | $94.5 \pm 0.7$ | $76.1 \pm 0.6$ | $79.6 \pm 0.6$ |
| | $\theta_b = bck_{1,2,3,4}$ | $75.9 \pm 1.4$ | $73.2 \pm 0.5$ | $94.0 \pm 0.5$ | $70.1 \pm 3.0$ | $78.3 \pm 0.8$ |
| Correlation | $\theta_b = \mathbf{I}$ | $79.1 \pm 0.5$ | $74.8 \pm 1.7$ | $93.5 \pm 0.5$ | $74.0 \pm 1.4$ | $80.3 \pm 0.5$ |
| | $\theta_b = bck_1$ | $79.7 \pm 1.0$ | $76.1 \pm 0.9$ | $94.0 \pm 0.6$ | $73.4 \pm 1.4$ | $80.8 \pm 0.5$ |
| | $\theta_b = bck_{1,2}$ | $76.9 \pm 1.5$ | $74.7 \pm 1.1$ | $94.0 \pm 0.4$ | $75.4 \pm 0.9$ | $80.3 \pm 0.5$ |
| | $\theta_b = bck_{1,2,3}$ | $78.7 \pm 1.3$ | $73.2 \pm 1.5$ | $93.3 \pm 0.2$ | $71.1 \pm 2.9$ | $79.1 \pm 0.4$ |
| | $\theta_b = bck_{1,2,3,4}$ | $76.2 \pm 0.6$ | $72.8 \pm 1.2$ | $94.0 \pm 0.4$ | $66.5 \pm 0.8$ | $77.4 \pm 0.3$ |

Table 8: Evaluations of the early-branching structure with augmentation in the unseen domain from (Li et al., 2017) using the ResNet18 and ResNet50 implementations. Here $bck_{1,2,3,4}$ is the four layers in a standard ResNet implementation with $bck_4$ close to the classifier. We do not implement $\theta_b = \mathbf{I}$ because it cannot coexist with our augmentation strategy. Our method consistently performs the best with an early-branching structure when the augmentation is applied.

| | ResNet18 structure | | | | | ResNet50 structure | | | | |
|---|---|---|---|---|---|---|---|---|---|---|
| | art | cartoon | photo | sketch | avg | art | cartoon | photo | sketch | avg |
| ERM | 78.0 | 73.4 | 94.1 | 73.6 | 79.8 | 85.1 | 78.0 | 97.3 | 72.2 | 83.1 |
| $\theta_b = bck_1$ | 81.3 | 75.7 | 94.6 | 78.1 | 82.4 | 86.2 | 81.4 | 97.3 | 78.7 | 85.9 |
| $\theta_b = bck_{1,2}$ | 81.2 | 76.4 | 94.5 | 76.8 | 82.2 | 85.3 | 80.4 | 97.3 | 78.5 | 85.4 |
| $\theta_b = bck_{1,2,3}$ | 75.5 | 75.8 | 92.4 | 75.2 | 79.7 | 82.2 | 78.1 | 96.5 | 76.0 | 83.2 |
| $\theta_b = bck_{1,2,3,4}$ | 54.8 | 50.6 | 77.7 | 52.3 | 58.9 | 62.4 | 57.5 | 80.0 | 46.4 | 61.6 |

we do not implement the setting when $\theta_b = \mathbf{I}$, because it can not coexist with our augmentation strategy. As shown in Table 8, the best performances can be achieved when $\theta_b = bck_1$ in both the two networks. The results suggest that our hypothesis is true in both situations when our augmentation strategy is applied or disabled.

### B.4 USING THE SAME AMOUNT OF PARAMETERS FOR THE EARLY-BRANCHING STRUCTURE

As the number of parameters for the overall training system is larger when sharing fewer blocks, one may wonder if it can affect the performances. To answer this question, we add extra blocks for the domain-estimation branch (i.e. $\theta_d$) to ensure that the training pipeline shares the same amount of parameters for different branching locations. Note we do not add parameters in the sharing part and the main classification branch (i.e. $\theta_b$ and $\theta_c$) because it will use more parameters than the baseline ERM model during the test phase, which is an unfair comparison. We conduct ablation studies on the PACS dataset (Li et al., 2017) and list the results in Table 9. The observations are in line with the findings in our manuscript: the early-branching structure still performs better than the common practice with lightweight prediction heads when using the same amount of parameters.

## C FURTHER RESULTS AND ANALYSIS

### C.1 RESULTS USING THE RESNET50 BACKBONE

To comprehensively evaluate the effectiveness of our method, we also conduct experiments using the ResNet50 backbone (He et al., 2016). In these experiments, we train the basic form of our method (i.e. ERM + HSIC), the strong baseline (i.e. ERM method (Vapnik, 1999)), and our method using the DomainBed benchmark (Gulrajani & Lopez-Paz, 2021), and we cite the results from citekim2021selfreg for other compared models.

Table 9: Evaluations of the early-branching structure in the unseen domain from (Li et al., 2017) using the same amount of parameters during the training phase. Here $bck_{1,2,3,4}$ is the four blocks in a standard ResNet implementation with $bck_4$ close to the classifier. We add extra blocks for the domain-estimation branch (i.e. $\theta_d$) to ensure that the training pipeline shares the same amount of parameters for different branching location. We do not add parameters for the sharing part and the main classification branch (i.e. $\theta_b$ and $\theta_c$) to ensure fair comparisons with the ERM method.

| | art | cartoon | photo | sketch | avg |
|---|---|---|---|---|---|
| ERM | $78.0 \pm 1.3$ | $73.4 \pm 0.8$ | $94.1 \pm 0.4$ | $73.6 \pm 2.2$ | $79.8 \pm 0.4$ |
| $\theta_b = \mathbf{I}$ | $79.1 \pm 0.7$ | $74.1 \pm 1.1$ | $94.6 \pm 0.4$ | $74.8 \pm 0.6$ | $80.7 \pm 0.5$ |
| $\theta_b = bck_1$ | $80.5 \pm 1.4$ | $74.1 \pm 0.9$ | $94.2 \pm 0.5$ | $75.6 \pm 0.3$ | $81.1 \pm 0.5$ |
| $\theta_b = bck_{1,2}$ | $78.5 \pm 1.1$ | $74.0 \pm 0.7$ | $94.4 \pm 0.4$ | $75.7 \pm 1.2$ | $80.6 \pm 0.4$ |
| $\theta_b = bck_{1,2,3}$ | $79.2 \pm 0.7$ | $72.4 \pm 2.1$ | $93.7 \pm 0.5$ | $75.0 \pm 1.2$ | $80.1 \pm 0.6$ |
| $\theta_b = bck_{1,2,3,4}$ | $76.0 \pm 0.9$ | $71.6 \pm 0.7$ | $93.5 \pm 0.3$ | $71.6 \pm 0.6$ | $78.2 \pm 0.5$ |

Table 10: Average accuracies on the DomainBed (Gulrajani & Lopez-Paz, 2021) benchmark using the ResNet50 (He et al., 2016) backbones. ERM + HSIC is our basic form without the augmentation and with $\theta_b = blk_1$. Results without † are directly cited from a previous work (Kim et al., 2021).

| | PACS | VLCS | OfficeHome | TerraInc | DomainNet | Average |
|---|---|---|---|---|---|---|
| ERM (Vapnik, 1999) | $85.5 \pm 0.2$ | $77.5 \pm 0.4$ | $66.5 \pm 0.3$ | $46.1 \pm 1.8$ | $40.9 \pm 0.1$ | 63.3 |
| IRM (Arjovsky et al., 2019) | $83.5 \pm 0.8$ | $78.5 \pm 0.5$ | $64.3 \pm 2.2$ | $47.6 \pm 0.8$ | $33.9 \pm 2.8$ | 61.2 |
| GroupGRO (Sagawa et al., 2020) | $84.4 \pm 0.8$ | $76.7 \pm 0.6$ | $66.0 \pm 0.7$ | $43.2 \pm 1.1$ | $33.3 \pm 0.2$ | 60.7 |
| Mixup (Yan et al., 2020) | $84.6 \pm 0.6$ | $77.4 \pm 0.6$ | $68.1 \pm 0.3$ | $47.9 \pm 0.8$ | $39.2 \pm 0.1$ | 63.4 |
| MLDG (Li et al., 2018a) | $84.9 \pm 1.0$ | $77.2 \pm 0.4$ | $66.8 \pm 0.6$ | $47.7 \pm 0.9$ | $41.2 \pm 0.1$ | 63.6 |
| CORAL (Sun & Saenko, 2016) | $86.2 \pm 0.3$ | $78.8 \pm 0.6$ | $68.7 \pm 0.3$ | $47.6 \pm 1.0$ | $41.5 \pm 0.1$ | 64.6 |
| MMD (Li et al., 2018b) | $84.6 \pm 0.5$ | $77.5 \pm 0.9$ | $66.3 \pm 0.1$ | $42.2 \pm 1.6$ | $23.4 \pm 9.5$ | 58.8 |
| DANN (Ganin et al., 2016) | $83.6 \pm 0.4$ | $78.6 \pm 0.4$ | $65.9 \pm 0.6$ | $46.7 \pm 0.5$ | $38.3 \pm 0.1$ | 62.6 |
| CDANN (Li et al., 2018c) | $82.6 \pm 0.9$ | $77.5 \pm 0.1$ | $65.8 \pm 1.3$ | $45.8 \pm 1.6$ | $38.3 \pm 0.3$ | 62.0 |
| MTL (Blanchard et al., 2017) | $84.6 \pm 0.5$ | $77.2 \pm 0.4$ | $66.4 \pm 0.5$ | $45.6 \pm 1.2$ | $40.6 \pm 0.1$ | 62.9 |
| SagNet (Nam et al., 2021) | $86.3 \pm 0.2$ | $77.8 \pm 0.5$ | $68.1 \pm 0.1$ | $48.6 \pm 1.0$ | $40.3 \pm 0.1$ | 64.2 |
| ARM (Zhang et al., 2020) | $85.1 \pm 0.4$ | $77.6 \pm 0.3$ | $64.8 \pm 0.3$ | $45.5 \pm 0.3$ | $35.5 \pm 0.2$ | 61.7 |
| VREx Krueger et al. (2021) | $84.9 \pm 0.6$ | $78.3 \pm 0.2$ | $66.4 \pm 0.6$ | $46.4 \pm 0.6$ | $33.6 \pm 2.9$ | 61.9 |
| RSC (Huang et al., 2020) | $85.2 \pm 0.9$ | $77.1 \pm 0.5$ | $65.5 \pm 0.9$ | $46.6 \pm 1.0$ | $38.9 \pm 0.5$ | 62.7 |
| SelfReg (Kim et al., 2021) | $85.6 \pm 0.4$ | $77.8 \pm 0.9$ | $67.9 \pm 0.7$ | $47.0 \pm 0.3$ | $42.8 \pm 0.0$ | 64.2 |
| ERM† (Vapnik, 1999) | $83.1 \pm 0.9$ | $77.7 \pm 0.8$ | $65.8 \pm 0.3$ | $46.5 \pm 0.9$ | $40.8 \pm 0.2$ | 62.8 |
| ERM + HSIC† | $84.7 \pm 0.5$ | $77.8 \pm 0.8$ | $67.3 \pm 0.4$ | $45.4 \pm 0.4$ | $40.6 \pm 0.1$ | 63.2 |
| Ours† | $85.9 \pm 0.4$ | $78.8 \pm 0.4$ | $68.4 \pm 0.3$ | $45.7 \pm 1.4$ | $40.5 \pm 0.4$ | 63.9 |

Results are listed in Table 10. We note that the independent constraint consistently improves the generalization performance as the basic form outperforms the baseline in most cases, and our method also obtains better performance than the baseline model and the basic form. These results illustrate the effectiveness of the independent constraint and the proposed augmentation strategy. Meanwhile, our method surpasses the baseline ERM model by 1.1 on average, while the reported top 3 methods (i.e. (Sun & Saenko, 2016), (Nam et al., 2021), (Kim et al., 2021)) lead their ERM model by 1.3, 0.9 and 0.9 in average. These results indicate that our method can also perform favorably against existing arts when implemented with a ResNet50 pipeline.

## C.2    Effectiveness of the MLP layer

We borrow the idea of using an additional MLP layer to prevent representation collapse from (Grill et al., 2020). Here we conduct an ablation study to examine if it is really necessary. To this end, we compare our method against a variant where the MLP layer is disabled in Eq. (3) (i.e. Ours w/o MLP in $\mathcal{L}_{sen}$). As listed in the first row in Table 11, integrating the MLP layer can bring improvements in most test domains, validating its effectiveness in improving the DG performance.

## C.3    Effectiveness of the sub loss terms in $\mathcal{L}_{sen}$

In the manuscript, we expect the target branch to be insensitive and the domain branch to be sensitive regarding the augmentation operation. Thus the original and augmented features are encouraged to

Table 11: Ablation studies regarding the effectiveness of the adopted MLP layer in $\mathcal{L}_{sen}$, and two sub loss terms in $\mathcal{L}_{sen}$ (i.e. $\mathcal{L}_{sim}$ and $\mathcal{L}_{dif}$). Experiments are conducted in PACS (Li et al., 2017) with the leave-one-out training-test strategy.

| model | art | cartoon | photo | sketch | avg |
|---|---|---|---|---|---|
| Ours w/o MLP in $\mathcal{L}_{sen}$ | $80.5 \pm 0.4$ | $75.1 \pm 0.6$ | $94.2 \pm 0.3$ | $77.7 \pm 1.3$ | $81.9 \pm 0.4$ |
| Ours w/o $\mathcal{L}_{sim}$ and $\mathcal{L}_{dif}$ | $79.3 \pm 0.4$ | $75.4 \pm 0.9$ | $94.5 \pm 0.2$ | $78.3 \pm 1.7$ | $81.9 \pm 0.5$ |
| Ours w/o $\mathcal{L}_{dif}$ | $81.2 \pm 0.5$ | $74.9 \pm 0.6$ | $94.6 \pm 0.4$ | $77.7 \pm 1.0$ | $82.1 \pm 0.7$ |
| Ours w/o $\mathcal{L}_{sim}$ | $80.8 \pm 0.6$ | $76.0 \pm 0.4$ | $94.1 \pm 0.2$ | $77.9 \pm 1.4$ | $82.2 \pm 0.4$ |
| Ours | $81.3 \pm 1.4$ | $75.7 \pm 0.2$ | $94.6 \pm 0.3$ | $78.1 \pm 0.7$ | $82.4 \pm 0.4$ |

Table 12: Evaluations regarding different hyper-parameter (i.e. $\alpha$ and $\beta$ in Eq. (5)) settings. We fix one parameter and tune another when conducting the experiments which are examined in PACS (Li et al., 2017) with the leave-one-out training-test strategy.

| hyper-parameters | | art | cartoon | photo | sketch | avg |
|---|---|---|---|---|---|---|
| $\beta = 1$ | $\alpha = 0.03$ | $80.7 \pm 0.7$ | $75.7 \pm 0.9$ | $95.1 \pm 0.2$ | $77.8 \pm 0.7$ | $82.3 \pm 0.4$ |
| | $\alpha = 0.3$ | $81.5 \pm 0.1$ | $76.3 \pm 1.2$ | $94.5 \pm 0.2$ | $78.6 \pm 1.0$ | $82.7 \pm 0.4$ |
| | $\alpha = 3$ | $79.2 \pm 0.8$ | $76.1 \pm 0.8$ | $92.9 \pm 0.1$ | $73.4 \pm 0.8$ | $80.4 \pm 0.3$ |
| | $\alpha = 30$ | $58.6 \pm 6.9$ | $45.3 \pm 9.0$ | $35.3 \pm 5.7$ | $30.3 \pm 4.5$ | $42.4 \pm 5.9$ |
| $\alpha = 0.3$ | $\beta = 0.1$ | $82.6 \pm 0.7$ | $76.3 \pm 0.3$ | $94.2 \pm 0.4$ | $74.9 \pm 0.7$ | $82.0 \pm 0.2$ |
| | $\beta = 1$ | $81.5 \pm 0.1$ | $76.3 \pm 1.2$ | $94.5 \pm 0.2$ | $78.6 \pm 1.0$ | $82.7 \pm 0.4$ |
| | $\beta = 10$ | $80.5 \pm 0.6$ | $75.5 \pm 0.2$ | $94.6 \pm 0.3$ | $76.1 \pm 0.9$ | $81.7 \pm 0.4$ |
| | $\beta = 100$ | $81.4 \pm 0.8$ | $76.8 \pm 0.7$ | $93.1 \pm 0.4$ | $75.9 \pm 0.6$ | $81.8 \pm 0.2$ |
| | $\beta = 1000$ | $53.1 \pm 7.2$ | $58.7 \pm 8.2$ | $63.3 \pm 11.9$ | $62.6 \pm 6.4$ | $59.4 \pm 8.4$ |

be similar in the target branch and different in the domain branch. Denoting the two sub loss terms as $\mathcal{L}_{sim}$ and $\mathcal{L}_{dif}$, we have,

$$\mathcal{L}_{sen} = \mathcal{L}_{sim} + \mathcal{L}_{dif}, \tag{10}$$

where $\mathcal{L}_{sim} = \|\theta_c(f) - \text{MLP}(\theta_c(\mathcal{A}(f)))\|_1 + \|\text{MLP}(\theta_c(f)) - \theta_c(\mathcal{A}(f))\|_1$, and $\mathcal{L}_{dif} = [\|\theta_d(f) - \text{MLP}(\theta_d(\mathcal{A}(f)))\|_1 - n]_- + [\|\text{MLP}(\theta_d(f)) - \theta_d(\mathcal{A}(f))\|_1 - n]_-$.

To evaluate the effectiveness of $\mathcal{L}_{sim}$ and $\mathcal{L}_{dif}$, we conduct experiments in the PACS dataset by disabling one and keeping another in $\mathcal{L}_{sen}$. Results are shown in Table 11. We observe that either one of the two terms can improve the performance, and the model performs the best when both two terms are integrated. These results indicate the effectiveness of the two sub loss terms.

### C.4 SENSITIVE TO THE HYPER-PARAMETERS.

Our overall algorithm contains two hyper-parameters (i.e. $\alpha$ and $\beta$ in Eq. (5)), which are empirically fixed as $\alpha = 0.3$ and $\beta = 1$. To analyze the sensitivity of our model to these hyper-parameters, we conduct ablation studies by evaluating our model on PACS (Li et al., 2017) using different settings of them. Note we fix the value for one hyper-parameter when analyzing another. Results are listed in Table 12. We observe that our method performs consistency well when the hyper-parameters are set to $\alpha \in [0.03, 0.3]$ and $\beta \in [0.1, 1]$.

### C.5 MORE COMPARISONS WITH EXISTING AUGMENTATION STRATEGIES

This section further compares our RDS augmentation strategy with other two methods that based on AdaIN (Huang & Belongie, 2017) (i.e. including MixStyle (Zhou et al., 2021) and DSU (Li et al., 2022)). First, for MixStyle, given a batch of training sample $x$, the new style statistics $\hat{\mu}_{mix}$ and $\hat{\sigma}_{mix}$ can be obtained by mixing the statistics of the original sample and their corresponding shuffled one:

$$\hat{\mu}_{mix} = \lambda\mu + (1 - \lambda)\tilde{\mu}, \text{ s.t. } \lambda \in \text{Beta}(\omega, \omega), \ \omega \in (0, \infty), \tag{11}$$

where $\text{Beta}(\cdot)$ denotes the Beta distribution, $\mu$ and $\tilde{\mu}$ compute the mean of the original batch and its shuffled version. A similar method is applied to obtain another style statistic $\hat{\sigma}_{mix}$. Second, for DSU,

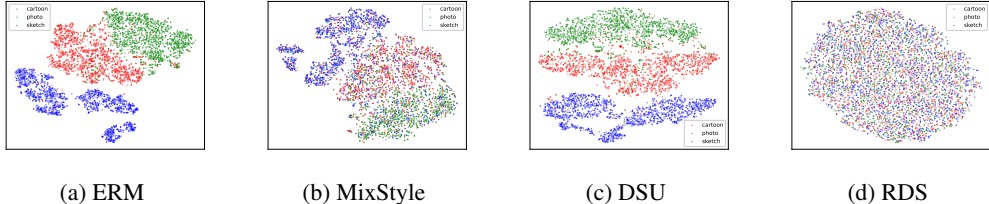

| (a) ERM | (b) MixStyle | (c) DSU | (d) RDS |

Figure 4: Visualizations of the original and augmented style statistics (concatenation of mean and variance). The PACS dataset Li et al. (2017) is used with art as the unseen domain during training. Augmented style statistics from MixStyle (Zhou et al., 2021) and DSU (Li et al., 2022) can still reveal domain information, while our RDS can create diverse and inhomogeneous statistics.

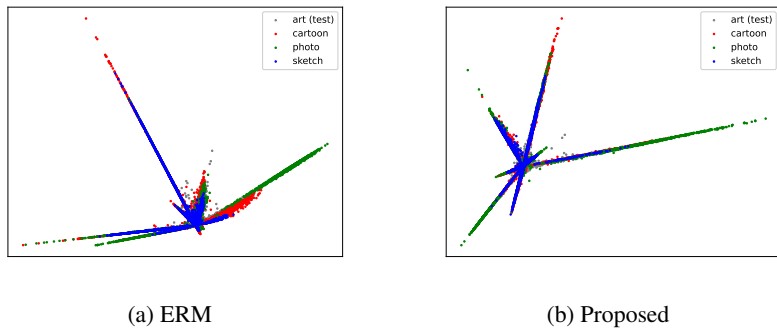

| (a) ERM | (b) Proposed |

Figure 5: Directly plot the 2d features from the penultimate layer. The PACS dataset (Li et al., 2017) is used with art as the unseen target domain. The 2d features are rather close for both two methods, indicating that they are less representative compared to the original 512d version.

their new statistics $\hat{\mu}_{dsu}$ and $\hat{\sigma}_{dsu}$ are obtained by adding perturbations to the original statistics:

$$\hat{\mu}_{dsu} = \mu + \phi\Sigma_\mu, \ \ \text{s.t.} \ \ \phi \in \mathcal{N}(0, 1), \tag{12}$$

where $\Sigma_\mu$ is the variance of $\mu$ (similar to our definition in Eq. (6)). The same approach is applied to obtain $\hat{\sigma}_{dsu}$.

To better illustrate the differences between their approaches and RDS, we plot the original and augmented style statistics in Figure 4. As shown in Figure 4 (b) and (c), style statistics augmented from MixStyle and DSU can still reveal the domain information, indicating that their augmentations may create homogeneous features. In contrast, statistics from RDS are diverse and do not contain domain information, demonstrating that RDS obtains unlimited inhomogeneous features. These visualizations and the ablation studies in Table 3 can validate the effectiveness of our RDS against other augmentation strategies.

## C.6 DIRECTLY PLOTTING THE 2D FEATURES

This section provides an alternate for visualizations of the features provided in Figure 2. Instead of using t-SNE to reduce the dimension of representation from the penultimate layer, we include an extra MLP layer to reduce the dimension into 2 directly and plot the results in Figure 5. Compared to the visualizations in Figure 2, we observe that the features in Figure 5 are close, and it is rather hard to distinguish the class and domain information for both the ERM and our method, indicating that the features from the penultimate layer may contain similar information. We believe the main reason is that it will cost a lot of useful information to reduce the dimension of the representation from the penultimate layer into 2 in our multi-object classification task.

# D DETAILED RESULTS OF THE BENCHMARK DATASETS

In this section, we present the average accuracy in each domain from different datasets. As shown in Table 13, 14, 15, 16, and 17, these results are detailed illustrations of the results in Table 2 in our manuscript. For all the experiments, we use the "training-domain validate set" as the model selection method. A total of 22 methods are examined for 60 trials in each unseen domain, and all methods are trained with the leave-one-out strategy using the ResNet18 (He et al., 2016) backbones.

Table 13: Average accuracies on the PACS (Li et al., 2017) datasets using the default hyper-parameter settings in DomainBed (Gulrajani & Lopez-Paz, 2021). Time (min) here is the training time for one trial per GPU for the corresponding method.

|  | art | cartoon | photo | sketch | Average | Time |
|---|---|---|---|---|---|---|
| ERM (Vapnik, 1999) | 78.0 ± 1.3 | 73.4 ± 0.8 | 94.1 ± 0.4 | 73.6 ± 2.2 | 79.8 ± 0.4 | 24 |
| ERM + HSIC | 79.3 ± 0.9 | 74.1 ± 2.0 | 94.8 ± 0.5 | 75.8 ± 1.6 | 81.0 ± 0.6 | 25 |
| IRM (Arjovsky et al., 2019) | 76.9 ± 2.6 | 75.1 ± 0.7 | 94.3 ± 0.4 | 77.4 ± 0.4 | 80.9 ± 0.5 | 18 |
| GroupGRO (Sagawa et al., 2020) | 77.7 ± 2.6 | 76.4 ± 0.3 | 94.0 ± 0.3 | 74.8 ± 1.3 | 80.7 ± 0.4 | 24 |
| Mixup (Yan et al., 2020) | 79.3 ± 1.1 | 74.2 ± 0.3 | 94.9 ± 0.3 | 68.3 ± 2.7 | 79.2 ± 0.9 | 18 |
| MLDG (Li et al., 2018a) | 78.4 ± 0.7 | 75.1 ± 0.5 | 94.8 ± 0.4 | 76.7 ± 0.8 | 81.3 ± 0.2 | 32 |
| CORAL (Sun & Saenko, 2016) | 81.5 ± 0.5 | 75.4 ± 0.7 | 95.2 ± 0.5 | 74.8 ± 0.4 | 81.7 ± 0.0 | 24 |
| MMD (Li et al., 2018b) | 81.3 ± 0.6 | 75.5 ± 1.0 | 94.0 ± 0.5 | 74.3 ± 1.5 | 81.3 ± 0.8 | 18 |
| DANN (Ganin et al., 2016) | 79.0 ± 0.6 | 72.5 ± 0.7 | 94.4 ± 0.5 | 70.8 ± 3.0 | 79.2 ± 0.3 | 17 |
| CDANN (Li et al., 2018c) | 80.4 ± 0.8 | 73.7 ± 0.3 | 93.1 ± 0.6 | 74.2 ± 1.7 | 80.3 ± 0.5 | 24 |
| MTL (Blanchard et al., 2017) | 78.7 ± 0.6 | 73.4 ± 1.0 | 94.1 ± 0.6 | 74.4 ± 3.0 | 80.1 ± 0.8 | 18 |
| SagNet (Nam et al., 2021) | 82.9 ± 0.4 | 73.2 ± 1.1 | 94.6 ± 0.5 | 76.1 ± 1.8 | 81.7 ± 0.6 | 24 |
| ARM (Zhang et al., 2020) | 79.4 ± 0.6 | 75.0 ± 0.7 | 94.3 ± 0.6 | 73.8 ± 0.6 | 80.6 ± 0.5 | 24 |
| VREx Krueger et al. (2021) | 74.4 ± 0.7 | 75.0 ± 0.4 | 93.3 ± 0.3 | 78.1 ± 0.9 | 80.2 ± 0.5 | 17 |
| RSC (Huang et al., 2020) | 78.5 ± 1.1 | 73.3 ± 0.9 | 93.6 ± 0.6 | 76.5 ± 1.4 | 80.5 ± 0.2 | 25 |
| SelfReg (Kim et al., 2021) | 82.5 ± 0.8 | 74.4 ± 1.5 | 95.4 ± 0.5 | 74.9 ± 1.3 | 81.8 ± 0.3 | 25 |
| MixStyle (Zhou et al., 2021) | 82.6 ± 1.2 | 76.3 ± 0.4 | 94.2 ± 0.3 | 77.5 ± 1.3 | 82.6 ± 0.4 | 25 |
| Fish (Shi et al., 2021) | 80.9 ± 1.0 | 75.9 ± 0.4 | 95.0 ± 0.4 | 76.2 ± 1.0 | 82.0 ± 0.3 | 52 |
| SD (Pezeshki et al., 2021) | 83.2 ± 0.6 | 74.6 ± 0.3 | 94.6 ± 0.1 | 75.1 ± 1.6 | 81.9 ± 0.3 | 25 |
| CAD (Ruan et al., 2022) | 83.9 ± 0.8 | 74.2 ± 0.6 | 94.6 ± 0.4 | 75.0 ± 1.2 | 81.9 ± 0.3 | 24 |
| CondCAD (Ruan et al., 2022) | 79.7 ± 1.0 | 74.2 ± 0.9 | 94.6 ± 0.4 | 74.8 ± 1.4 | 80.8 ± 0.5 | 26 |
| Fishr (Rame et al., 2022) | 81.2 ± 0.4 | 75.8 ± 0.8 | 94.3 ± 0.3 | 73.8 ± 0.6 | 81.3 ± 0.3 | 17 |
| Ours | 81.3 ± 1.4 | 75.7 ± 0.2 | 94.6 ± 0.3 | 78.1 ± 0.7 | 82.4 ± 0.4 | 29 |

Table 14: Average accuracies on the VLCS (Fang et al., 2013) datasets using the default hyper-parameter settings in DomainBed (Gulrajani & Lopez-Paz, 2021).

|  | Caltech | LabelMe | Sun | VOC | Average |
|---|---|---|---|---|---|
| ERM (Vapnik, 1999) | 97.7 ± 0.3 | 62.1 ± 0.9 | 70.3 ± 0.9 | 73.2 ± 0.7 | 75.8 ± 0.2 |
| ERM + HSIC | 97.0 ± 0.3 | 62.3 ± 0.3 | 70.1 ± 0.1 | 75.1 ± 0.7 | 76.1 ± 0.1 |
| IRM (Arjovsky et al., 2019) | 96.1 ± 0.8 | 62.5 ± 0.3 | 69.9 ± 0.7 | 72.0 ± 1.4 | 75.1 ± 0.1 |
| GroupGRO (Sagawa et al., 2020) | 96.7 ± 0.6 | 61.7 ± 1.5 | 70.2 ± 1.8 | 72.9 ± 0.6 | 75.4 ± 1.0 |
| Mixup (Yan et al., 2020) | 95.6 ± 1.5 | 62.7 ± 0.4 | 71.3 ± 0.3 | 75.4 ± 0.2 | 76.2 ± 0.3 |
| MLDG (Li et al., 2018a) | 95.8 ± 0.5 | 63.3 ± 0.8 | 68.5 ± 0.5 | 73.1 ± 0.8 | 75.2 ± 0.3 |
| CORAL (Sun & Saenko, 2016) | 96.5 ± 0.3 | 63.0 ± 0.1 | 69.1 ± 0.6 | 73.8 ± 1.0 | 75.5 ± 0.4 |
| MMD (Li et al., 2018b) | 96.0 ± 0.8 | 64.3 ± 0.6 | 68.5 ± 0.6 | 70.8 ± 0.1 | 74.9 ± 0.5 |
| DANN (Ganin et al., 2016) | 97.2 ± 0.1 | 63.3 ± 0.6 | 70.2 ± 0.9 | 74.4 ± 0.2 | 76.3 ± 0.2 |
| CDANN (Li et al., 2018c) | 95.4 ± 1.2 | 62.6 ± 0.6 | 69.9 ± 1.3 | 76.2 ± 0.5 | 76.0 ± 0.5 |
| MTL (Blanchard et al., 2017) | 94.4 ± 2.3 | 65.0 ± 0.6 | 69.6 ± 0.6 | 71.7 ± 1.3 | 75.2 ± 0.3 |
| SagNet (Nam et al., 2021) | 94.9 ± 0.7 | 61.9 ± 0.7 | 69.6 ± 1.3 | 75.2 ± 0.6 | 75.4 ± 0.8 |
| ARM (Zhang et al., 2020) | 96.9 ± 0.5 | 61.9 ± 0.4 | 71.6 ± 0.1 | 73.3 ± 0.4 | 75.9 ± 0.3 |
| VREx Krueger et al. (2021) | 96.2 ± 0.0 | 62.5 ± 1.3 | 69.3 ± 0.9 | 73.1 ± 1.2 | 75.3 ± 0.6 |
| RSC (Huang et al., 2020) | 96.2 ± 0.0 | 63.6 ± 1.3 | 69.8 ± 1.0 | 72.0 ± 0.4 | 75.4 ± 0.3 |
| SelfReg (Kim et al., 2021) | 95.8 ± 0.6 | 63.4 ± 1.1 | 71.1 ± 0.6 | 75.3 ± 0.6 | 76.4 ± 0.7 |
| MixStyle (Zhou et al., 2021) | 97.3 ± 0.3 | 61.6 ± 0.1 | 70.4 ± 0.7 | 71.3 ± 1.9 | 75.2 ± 0.7 |
| Fish (Shi et al., 2021) | 97.4 ± 0.2 | 63.4 ± 0.1 | 71.5 ± 0.4 | 75.2 ± 0.7 | 76.9 ± 0.2 |
| SD (Pezeshki et al., 2021) | 96.5 ± 0.4 | 62.2 ± 0.0 | 69.7 ± 0.9 | 73.6 ± 0.4 | 75.5 ± 0.4 |
| CAD (Ruan et al., 2022) | 94.5 ± 0.9 | 63.5 ± 0.6 | 70.4 ± 1.2 | 72.4 ± 1.3 | 75.2 ± 0.6 |
| CondCAD (Ruan et al., 2022) | 96.5 ± 0.8 | 62.6 ± 0.4 | 69.1 ± 0.2 | 76.0 ± 0.2 | 76.1 ± 0.3 |
| Fishr (Rame et al., 2022) | 97.2 ± 0.6 | 63.3 ± 0.7 | 70.4 ± 0.6 | 74.0 ± 0.8 | 76.2 ± 0.3 |
| Ours | 97.0 ± 0.2 | 63.7 ± 0.9 | 71.0 ± 0.3 | 74.4 ± 0.8 | 76.5 ± 0.4 |

Table 15: Average accuracies on the OfficeHome (Venkateswara et al., 2017) datasets using the default hyper-parameter settings in DomainBed (Gulrajani & Lopez-Paz, 2021).

| | art | clipart | product | real | Average |
|---|---|---|---|---|---|
| ERM (Vapnik, 1999) | 52.2 ± 0.2 | 48.7 ± 0.5 | 69.9 ± 0.5 | 71.7 ± 0.5 | 60.6 ± 0.2 |
| ERM + HSIC | 53.1 ± 0.8 | 47.8 ± 1.1 | 70.3 ± 0.8 | 71.4 ± 0.7 | 60.7 ± 0.3 |
| IRM (Arjovsky et al., 2019) | 49.7 ± 0.2 | 46.8 ± 0.5 | 67.5 ± 0.4 | 68.1 ± 0.6 | 58.0 ± 0.1 |
| GroupGRO (Sagawa et al., 2020) | 52.6 ± 1.1 | 48.2 ± 0.9 | 69.9 ± 0.4 | 71.5 ± 0.8 | 60.6 ± 0.3 |
| Mixup (Yan et al., 2020) | 54.0 ± 0.7 | 49.3 ± 0.7 | 70.7 ± 0.7 | 72.6 ± 0.3 | 61.7 ± 0.5 |
| MLDG (Li et al., 2018a) | 53.1 ± 0.3 | 48.4 ± 0.3 | 70.5 ± 0.7 | 71.7 ± 0.4 | 60.9 ± 0.2 |
| CORAL (Sun & Saenko, 2016) | 55.1 ± 0.7 | 49.7 ± 0.9 | 71.8 ± 0.2 | 73.1 ± 0.5 | 62.4 ± 0.4 |
| MMD (Li et al., 2018b) | 50.9 ± 1.0 | 48.7 ± 0.3 | 69.3 ± 0.7 | 70.7 ± 1.3 | 59.9 ± 0.4 |
| DANN (Ganin et al., 2016) | 51.8 ± 0.5 | 47.1 ± 0.1 | 69.1 ± 0.7 | 70.2 ± 0.7 | 59.5 ± 0.5 |
| CDANN (Li et al., 2018c) | 51.4 ± 0.5 | 46.9 ± 0.6 | 68.4 ± 0.5 | 70.4 ± 0.4 | 59.3 ± 0.4 |
| MTL (Blanchard et al., 2017) | 51.6 ± 1.5 | 47.7 ± 0.5 | 69.1 ± 0.3 | 71.0 ± 0.6 | 59.9 ± 0.5 |
| SagNet (Nam et al., 2021) | 55.3 ± 0.4 | 49.6 ± 0.2 | 72.1 ± 0.4 | 73.2 ± 0.4 | 62.5 ± 0.3 |
| ARM (Zhang et al., 2020) | 51.3 ± 0.9 | 48.5 ± 0.4 | 68.0 ± 0.3 | 70.6 ± 0.1 | 59.6 ± 0.3 |
| VREx Krueger et al. (2021) | 51.1 ± 0.3 | 47.4 ± 0.6 | 69.0 ± 0.4 | 70.5 ± 0.4 | 59.5 ± 0.1 |
| RSC (Huang et al., 2020) | 49.0 ± 0.1 | 46.2 ± 1.5 | 67.8 ± 0.7 | 70.6 ± 0.3 | 58.4 ± 0.6 |
| SelfReg (Kim et al., 2021) | 55.1 ± 0.8 | 49.2 ± 0.6 | 72.2 ± 0.3 | 73.0 ± 0.3 | 62.4 ± 0.1 |
| MixStyle (Zhou et al., 2021) | 50.8 ± 0.6 | 51.4 ± 1.1 | 67.6 ± 1.3 | 68.8 ± 0.5 | 59.6 ± 0.8 |
| Fish (Shi et al., 2021) | 54.6 ± 1.0 | 49.6 ± 1.0 | 71.3 ± 0.6 | 72.4 ± 0.2 | 62.0 ± 0.6 |
| SD (Pezeshki et al., 2021) | 55.0 ± 0.4 | 51.3 ± 0.5 | 72.5 ± 0.2 | 72.7 ± 0.3 | 62.9 ± 0.2 |
| CAD (Ruan et al., 2022) | 52.1 ± 0.6 | 48.3 ± 0.5 | 69.7 ± 0.3 | 71.9 ± 0.4 | 60.5 ± 0.3 |
| CondCAD (Ruan et al., 2022) | 53.3 ± 0.6 | 48.4 ± 0.2 | 69.8 ± 0.9 | 72.6 ± 0.1 | 61.0 ± 0.4 |
| Fishr (Rame et al., 2022) | 52.6 ± 0.9 | 48.6 ± 0.3 | 69.9 ± 0.6 | 72.4 ± 0.4 | 60.9 ± 0.3 |
| Ours | 54.5 ± 0.3 | 52.4 ± 0.2 | 71.4 ± 0.5 | 70.3 ± 0.3 | 62.2 ± 0.1 |

Table 16: Average accuracies on the TerraInc (Beery et al., 2018) datasets using the default hyper-parameter settings in DomainBed (Gulrajani & Lopez-Paz, 2021).

| | L100 | L38 | L43 | L46 | Average |
|---|---|---|---|---|---|
| ERM (Vapnik, 1999) | 42.1 ± 2.5 | 30.1 ± 1.2 | 48.9 ± 0.6 | 34.0 ± 1.1 | 38.8 ± 1.0 |
| ERM + HSIC | 40.0 ± 2.6 | 32.4 ± 1.4 | 50.0 ± 0.4 | 32.9 ± 0.9 | 38.8 ± 1.1 |
| IRM (Arjovsky et al., 2019) | 41.8 ± 1.8 | 29.0 ± 3.6 | 49.6 ± 2.1 | 33.1 ± 1.5 | 38.4 ± 0.9 |
| GroupGRO (Sagawa et al., 2020) | 45.3 ± 4.6 | 36.1 ± 4.4 | 51.0 ± 0.8 | 33.7 ± 0.9 | 41.5 ± 2.0 |
| Mixup (Yan et al., 2020) | 49.4 ± 2.0 | 35.9 ± 1.8 | 53.0 ± 0.7 | 30.0 ± 0.9 | 42.1 ± 0.7 |
| MLDG (Li et al., 2018a) | 39.6 ± 2.3 | 33.2 ± 2.7 | 52.4 ± 0.5 | 35.1 ± 1.5 | 40.1 ± 0.9 |
| CORAL (Sun & Saenko, 2016) | 46.7 ± 3.2 | 36.9 ± 4.3 | 49.5 ± 1.9 | 32.5 ± 0.7 | 41.4 ± 1.8 |
| MMD (Li et al., 2018b) | 49.1 ± 1.2 | 36.4 ± 4.8 | 50.4 ± 2.1 | 32.3 ± 1.5 | 42.0 ± 1.0 |
| DANN (Ganin et al., 2016) | 44.3 ± 3.6 | 28.0 ± 1.5 | 47.9 ± 1.0 | 31.3 ± 0.6 | 37.9 ± 0.9 |
| CDANN (Li et al., 2018c) | 36.9 ± 6.4 | 32.7 ± 6.2 | 51.1 ± 1.3 | 33.5 ± 0.5 | 38.6 ± 2.3 |
| MTL (Blanchard et al., 2017) | 45.2 ± 2.6 | 31.0 ± 1.6 | 50.6 ± 1.1 | 34.9 ± 0.4 | 40.4 ± 1.0 |
| SagNet (Nam et al., 2021) | 36.3 ± 4.7 | 40.3 ± 2.0 | 52.5 ± 0.6 | 33.3 ± 1.3 | 40.6 ± 1.5 |
| ARM (Zhang et al., 2020) | 41.5 ± 4.5 | 27.7 ± 2.4 | 50.9 ± 1.0 | 29.6 ± 1.5 | 37.4 ± 1.9 |
| VREx Krueger et al. (2021) | 48.0 ± 1.7 | 41.1 ± 1.5 | 51.8 ± 1.5 | 32.0 ± 1.2 | 43.2 ± 0.3 |
| RSC (Huang et al., 2020) | 42.8 ± 2.4 | 32.2 ± 3.8 | 49.6 ± 0.9 | 32.9 ± 1.2 | 39.4 ± 1.3 |
| SelfReg (Kim et al., 2021) | 46.1 ± 1.5 | 34.5 ± 1.6 | 49.8 ± 0.3 | 34.7 ± 1.5 | 41.3 ± 0.3 |
| MixStyle (Zhou et al., 2021) | 50.6 ± 1.9 | 28.0 ± 4.5 | 52.1 ± 0.7 | 33.0 ± 0.2 | 40.9 ± 1.1 |
| Fish (Shi et al., 2021) | 46.3 ± 3.0 | 29.0 ± 1.1 | 52.7 ± 1.2 | 32.8 ± 1.0 | 40.2 ± 0.6 |
| SD (Pezeshki et al., 2021) | 45.5 ± 1.9 | 33.2 ± 3.1 | 52.9 ± 0.7 | 36.4 ± 0.8 | 42.0 ± 1.0 |
| CAD (Ruan et al., 2022) | 43.1 ± 2.6 | 31.1 ± 1.9 | 53.1 ± 1.6 | 34.7 ± 1.3 | 40.5 ± 0.4 |
| CondCAD (Ruan et al., 2022) | 44.4 ± 2.9 | 32.9 ± 2.5 | 50.5 ± 1.3 | 30.8 ± 0.5 | 39.7 ± 0.4 |
| Fishr (Rame et al., 2022) | 49.9 ± 3.3 | 36.6 ± 0.9 | 49.8 ± 0.2 | 34.2 ± 1.3 | 42.6 ± 1.0 |
| Ours | 50.8 ± 4.5 | 35.8 ± 0.9 | 51.1 ± 1.3 | 35.2 ± 2.6 | 43.2 ± 1.3 |

Table 17: Average accuracies on the DomainNet (Peng et al., 2019) datasets using the default hyper-parameter settings in DomainBed (Gulrajani & Lopez-Paz, 2021).

| | clip | info | paint | quick | real | sketch | Average |
|---|---|---|---|---|---|---|---|
| ERM (Vapnik, 1999) | $50.4 \pm 0.2$ | $14.0 \pm 0.2$ | $40.3 \pm 0.5$ | $11.7 \pm 0.2$ | $52.0 \pm 0.2$ | $43.2 \pm 0.3$ | $35.3 \pm 0.1$ |
| ERM + HSIC | $50.6 \pm 0.2$ | $14.2 \pm 0.1$ | $39.8 \pm 0.4$ | $12.6 \pm 0.0$ | $51.8 \pm 0.2$ | $43.1 \pm 0.3$ | $35.4 \pm 0.1$ |
| IRM (Arjovsky et al., 2019) | $43.2 \pm 0.9$ | $12.6 \pm 0.3$ | $35.0 \pm 1.4$ | $9.9 \pm 0.4$ | $43.4 \pm 3.0$ | $38.4 \pm 0.4$ | $30.4 \pm 1.0$ |
| GroupGRO (Sagawa et al., 2020) | $38.2 \pm 0.5$ | $13.0 \pm 0.3$ | $28.7 \pm 0.3$ | $8.2 \pm 0.1$ | $43.4 \pm 0.5$ | $33.7 \pm 0.0$ | $27.5 \pm 0.1$ |
| Mixup (Yan et al., 2020) | $48.9 \pm 0.3$ | $13.6 \pm 0.3$ | $39.5 \pm 0.5$ | $10.9 \pm 0.4$ | $49.9 \pm 0.2$ | $41.2 \pm 0.2$ | $34.0 \pm 0.0$ |
| MLDG (Li et al., 2018a) | $51.1 \pm 0.3$ | $14.1 \pm 0.3$ | $40.7 \pm 0.3$ | $11.7 \pm 0.1$ | $52.3 \pm 0.3$ | $42.7 \pm 0.2$ | $35.4 \pm 0.0$ |
| CORAL (Sun & Saenko, 2016) | $51.2 \pm 0.2$ | $15.4 \pm 0.2$ | $42.0 \pm 0.2$ | $12.7 \pm 0.1$ | $52.0 \pm 0.3$ | $43.4 \pm 0.0$ | $36.1 \pm 0.2$ |
| MMD (Li et al., 2018b) | $16.6 \pm 13.3$ | $0.3 \pm 0.0$ | $12.8 \pm 10.4$ | $0.3 \pm 0.0$ | $17.1 \pm 13.7$ | $0.4 \pm 0.0$ | $7.9 \pm 6.2$ |
| DANN (Ganin et al., 2016) | $45.0 \pm 0.2$ | $12.8 \pm 0.2$ | $36.0 \pm 0.2$ | $10.4 \pm 0.3$ | $46.7 \pm 0.3$ | $38.0 \pm 0.3$ | $31.5 \pm 0.1$ |
| CDANN (Li et al., 2018c) | $45.3 \pm 0.2$ | $12.6 \pm 0.2$ | $36.6 \pm 0.2$ | $10.3 \pm 0.4$ | $47.5 \pm 0.1$ | $38.9 \pm 0.4$ | $31.8 \pm 0.2$ |
| MTL (Blanchard et al., 2017) | $50.6 \pm 0.2$ | $14.0 \pm 0.4$ | $39.6 \pm 0.3$ | $12.0 \pm 0.3$ | $52.1 \pm 0.1$ | $41.5 \pm 0.0$ | $35.0 \pm 0.0$ |
| SagNet (Nam et al., 2021) | $51.0 \pm 0.1$ | $14.6 \pm 0.1$ | $40.2 \pm 0.2$ | $12.1 \pm 0.2$ | $51.5 \pm 0.3$ | $42.4 \pm 0.1$ | $35.3 \pm 0.1$ |
| ARM (Zhang et al., 2020) | $43.0 \pm 0.2$ | $11.7 \pm 0.2$ | $34.6 \pm 0.1$ | $9.8 \pm 0.4$ | $43.2 \pm 0.3$ | $37.0 \pm 0.3$ | $29.9 \pm 0.1$ |
| VREx Krueger et al. (2021) | $39.2 \pm 1.6$ | $11.9 \pm 0.4$ | $31.2 \pm 1.3$ | $10.2 \pm 0.4$ | $41.5 \pm 1.8$ | $34.8 \pm 0.8$ | $28.1 \pm 1.0$ |
| RSC (Huang et al., 2020) | $39.5 \pm 3.7$ | $11.4 \pm 0.8$ | $30.5 \pm 3.1$ | $10.2 \pm 0.8$ | $41.0 \pm 1.4$ | $34.7 \pm 2.6$ | $27.9 \pm 2.0$ |
| SelfReg (Kim et al., 2021) | $47.9 \pm 0.3$ | $15.1 \pm 0.3$ | $41.2 \pm 0.2$ | $11.7 \pm 0.3$ | $48.8 \pm 0.0$ | $43.8 \pm 0.3$ | $34.7 \pm 0.2$ |
| MixStyle (Zhou et al., 2021) | $49.1 \pm 0.4$ | $13.4 \pm 0.0$ | $39.3 \pm 0.0$ | $11.4 \pm 0.4$ | $47.7 \pm 0.3$ | $42.7 \pm 0.1$ | $33.9 \pm 0.1$ |
| Fish (Shi et al., 2021) | $51.5 \pm 0.3$ | $14.5 \pm 0.2$ | $40.4 \pm 0.3$ | $11.7 \pm 0.5$ | $52.6 \pm 0.2$ | $42.1 \pm 0.1$ | $35.5 \pm 0.0$ |
| SD (Pezeshki et al., 2021) | $51.3 \pm 0.3$ | $15.5 \pm 0.1$ | $41.5 \pm 0.3$ | $12.6 \pm 0.2$ | $52.9 \pm 0.2$ | $44.0 \pm 0.4$ | $36.3 \pm 0.2$ |
| CAD (Ruan et al., 2022) | $45.4 \pm 1.0$ | $12.1 \pm 0.5$ | $34.9 \pm 1.1$ | $10.2 \pm 0.6$ | $45.1 \pm 1.6$ | $38.5 \pm 0.6$ | $31.0 \pm 0.8$ |
| CondCAD (Ruan et al., 2022) | $46.1 \pm 1.0$ | $13.3 \pm 0.4$ | $36.1 \pm 1.4$ | $10.7 \pm 0.2$ | $46.8 \pm 1.3$ | $38.7 \pm 0.7$ | $31.9 \pm 0.7$ |
| Fishr (Rame et al., 2022) | $47.8 \pm 0.7$ | $14.6 \pm 0.2$ | $40.0 \pm 0.3$ | $11.9 \pm 0.2$ | $49.2 \pm 0.7$ | $41.7 \pm 0.1$ | $34.2 \pm 0.3$ |
| Ours | $48.8 \pm 0.7$ | $14.7 \pm 0.2$ | $40.8 \pm 0.1$ | $11.4 \pm 0.2$ | $49.1 \pm 0.1$ | $44.6 \pm 0.0$ | $34.9 \pm 0.1$ |

