# OpenReview forum: "Domain Generalization via Independent Regularization from Early-branching Networks"
_ICLR.cc/2023/Conference — Submitted to ICLR 2023_

### Official Review · Reviewer_sawe · 2022-10-19

**Confidence:** 5
**Correctness:** 4
**Technical Novelty And Significance:** 3
**Empirical Novelty And Significance:** 3
**Recommendation:** 8

**Clarity, Quality, Novelty And Reproducibility:**

**Clarity**: The paper is well-written and easy to follow.

**Quality**: The results provide some new insights regarding how to design a better two-branch model for learning disentangled representations.

**Novelty**: The design of the two-branch neural network is well-motivated with numerical results as back-up. The combination of the disentangled feature learning loss and the domain augmentation method is novel.

**Reproducibility**: The implementation detail has been given in the paper.

**Strength And Weaknesses:**

**Strengths**

First of all, the paper is well-written and well-organized, and both pros and cons of the proposed method are properly discussed in the paper. The idea of learning disentangled representations isn't new in domain generalization. But the study is novel and provides useful tips to the field. The entire framework—including the two-branch network, the correlation minimization loss, and the domain augmentation strategy—is well-motivated. The results clearly show that such a simple method works well in improving model generalization. The two measures, Top-5 and Score used in Table 5, deserve praise and could be of interest to the DomainBed community (where how to properly quantify progress remains an open question).

**Weaknesses**

Overall, there is no major issue that could lead to rejection of the paper. Only a few minor comments are given below, which the authors can use to further improve the paper.

1. The proposed domain augmentation strategy in 3.3 is quite similar to the following work, "Uncertainty Modeling for Out-of-Distribution Generalization" published in ICLR'22, which has been cited in the paper. The current discussions in the end of 3.3 aren't very convincing as both methods are based on random sampling. Could the authors point out the major differences, perhaps using math equations?

2. Have the authors tried other correlation minimization losses besides HSIC? Having more discussions about this could greatly help practitioners choose which loss to use.

3. The Augmentation part in Sec. 2 misses those work using generative modeling to achieve data augmentation, e.g., [1, 2].

4. The 2D t-SNE visualization isn't fully convincing because the spreading of the features could well be manipulated by using different parameters in the t-SNE method. Would it be possible to provide the following visualization: reduce the feature dimension to 2 in the last layer and plot the features directly on a 2D plane?

[1] Zhou, K., Yang, Y., Hospedales, T., & Xiang, T. (2020, August). Learning to generate novel domains for domain generalization. In European conference on computer vision (pp. 561-578). Springer, Cham.

[2] Carlucci, F. M., Russo, P., Tommasi, T., & Caputo, B. (2019, October). Hallucinating agnostic images to generalize across domains. In 2019 IEEE/CVF International Conference on Computer Vision Workshop (ICCVW) (pp. 3227-3234). IEEE.

**Summary Of The Paper:**

The paper presents an interesting study on disentangled representation learning for domain generalization. Technically, the main idea is to build a two-branch network, one for target classification while the other for domain classification. The paper uses carefully designed experiments to demonstrate that separating the two branches at an early stage of a neural network is more effective in learning disentangled representations.

To motivate the method design, the paper starts from a basic learning framework, which combines a classification loss with a correlation minimization loss (the latter is used for learning disentangled representations). Then, the paper motivates the use of domain augmentation (i.e., MixStyle) to facilitate the disentanglement, followed by the introduction of a modified domain augmentation strategy.

The main experiments are conducted on the DomainBed benchmark where the proposed method outperforms most baselines.

**Summary Of The Review:**

I recommend to accept the paper because the method is simple, well-motivated and well-evaluated. The findings could be useful to future work on learning disentangled representations for domain generalization.

== Post-rebuttal update ==

I have read the rebuttal as well as other reviewers' comments. The authors have done a good job in addressing the questions I raised in the first-round review. My view on the novelty and significance of the paper remains the same: the paper should be accepted.

---

> ### Author Response · Authors · 2022-11-16
> **Detailed discussion of related works and reducing the dimensions directly instead of using t-SNE for visualization**
>
> We appreciate the reviewer for the comments and we answer the raised questions below.
>
>
>
> * **1. Details of the augmentation strategy in DSU**
>
> Different from our RDS, augmented style statistics from DSU [3] are obtained by adding perturbations to the original statistics, and there is no actual sampling operation in their implementation. Given the style statistics ($\mu$ and $\sigma$) from the original sample, augmented style statistics $dsu_{\mu}$ and $dsu_{\sigma}$  from DSU can be represented as:
>
> $dsu_{\mu}$ = $\mu + \phi_{\mu}\Sigma_{\mu}$, s.t. $\phi_{\mu} \in \mathcal{N}(0,1)$
>
> $dsu_{\sigma}$ = $\sigma+ \phi_{\sigma}\Sigma_{\sigma}$, s.t. $\phi_{\sigma} \in \mathcal{N}(0,1)$
>
> where $\Sigma_{\mu}$ is the variance of $\mu$ (same as the definition in Eq. (6) in our manuscript), and similar for $\Sigma_{\sigma}$.
>
> Differently, augmented style statistics from our RDS (i.e. $\hat{\mu}$ and $\hat{\sigma}$) are sampled from the distributions of $\mathcal{N}(\mathop{\mathbb{E}}\nolimits_{\mu}, \Sigma_{\mu})$ and $\mathcal{N}(\mathop{\mathbb{E}}\nolimits_{\sigma}, \Sigma_{\sigma})$, and the sampling process can be further combined with the constraint in Eq. (7) to select diverse styles. Following your suggestion,  we also provide detailed visual comparisons of these two augmentation strategies in Figure 4 in the appendix, which shows that style statistics from DSU still contain domain information, indicating that their method may synthesize more homogeneous domain features compared with our RDS method. Please also refer to Section C.5 in the appendix for details.
>
>
>
> * **2. Other independent constraints besides HSIC**
>
> We also use other independent constraints, such as orthogonal and correlation minimization, to evaluate our method. Results are listed in the constraints part in Table 3, which shows that HSIC is a better choice compared with others.
>
>
>  * **3. Discuss related works**
>
>
> Thanks for the notification. We discuss the mentioned arts in our related works in the revised manuscript.
>
>
>
> * **4. Directly plotting  the 2d representations**
>
> We use the default parameters in t-SNE for dimension reduction before plotting, which is a common practice for feature visualization when confronting data with large dimensions. Following your suggestion, we include an extra MLP layer to reduce the dimension of representation from the penultimate layer into 2, which is 512 in the original ResNet18 backbone. We plot the results in Figure 5 in our appendix. Compared to the visualizations in Figure 2, we observe that the features are close, and it is rather hard to distinguish the class and domain information for both the ERM and our methods, indicating that the features from the penultimate layer may contain similar information. We believe the main reason is that it will cost much useful information to directly reduce the feature dimension into 2 in our multi-class classification task, and the suggested visualization method may be better suited for a binary classification task. Please also refer to Section C.6 in our appendix for details.
>
> $\textbf{Reference}$
>
> [3] Li, Xiaotong, et al. "Uncertainty modeling for out-of-distribution generalization". In ICLR, 2022.

---

> > ### Comment · Reviewer_sawe · 2022-11-19
> > **Good work**
> >
> > Thanks for the detailed responses. I have updated my review and keep the rating unchanged.

---

> > > ### Author Response · Authors · 2022-11-19
> > > **Thanks for your support**
> > >
> > > We would like to thank the reviewer for the positive comments on our work. We sincerely appreciate all your valuable comments and suggestions.

---

### Official Review · Reviewer_VzKA · 2022-10-23

**Confidence:** 4
**Correctness:** 4
**Technical Novelty And Significance:** 3
**Empirical Novelty And Significance:** 2
**Recommendation:** 6

**Clarity, Quality, Novelty And Reproducibility:**

This paper is well-structured and explains the method clearly. The method is clear and looks like easy to reproduce.

**Strength And Weaknesses:**

Pros:
1.	The logic of this paper is rigorous, and the structure is clear.
2.	The ablation study shows that the HSIC measurement is a better choice compared to the orthogonal constraint and the correlation constraint.
3.	The authors proposed a constrain to obtain inhomogeneous styles to synthesize new domain information.
4.	Experiments on benchmarks show the effectiveness of the proposed method.

Cons:
1.	The authors try to demonstrate early branch architecture has a relatively better performance. However, in the comparing process of different branching location in the network, the number of parameters of shared base feature extractor and two branches are always varies. Thus, such a comparison cannot eliminate the number of parameters’ influence on the performance.
2.	About augments ablation study in Table 3. When the photo or carton is the target domain, the performance of RDS with proposed sampling scheme has no noticeable improvement compared to naive sampling without Eq. (7) , which should be discussed.
3.	Some new works are not discussed in the paper, e.g., Style Neophile: Constantly Seeking Novel Styles for Domain Generalization (CVPR 2022).
4.	The best results in the experiment table should be in bold type to let others find the results easily. Also, the second-best results should be underlined.

**Summary Of The Paper:**

This paper tries to solve Domain Generalization (DG) classification problem following the simple idea of decoupling the domain-invariant feature and domain-specific feature by applying an early branching strategy, a new independence measurement HSIC, and a new feature-level data augmentation method generating samples of new domain. Comprehensive experiments on several DG classification benchmarks demonstrate the effectiveness of the proposed method.

**Summary Of The Review:**

The paper follows a simple idea has limited novelty. But the proposed methods make this simple idea more effective. Comprehensive experiments show the effectiveness of early branching and HISC measurement. This method brings new ideas or approaches to handle how to generate novel domain samples and combined it with previous method to form a feature level data augmentation method.

---

> ### Author Response · Authors · 2022-11-16
> **Experiments with the same number of parameters and including relevant discussions**
>
> We thank the reviewer for the comments and we answer the raised questions below.
>
>
>
> * **1. Experiments with the same number of parameters during training**
>
>  Despite the varying branching locations, our framework ensures that we use the same number of parameters as the baseline ERM method during the test phase. That is, we only use the base feature extractor $\theta_b$, the target branch $\theta_c$ and $\phi_c$ during the test phase, and the combination of these modules is kept the same with the structure of ERM, which ensures fair comparisons between different branching locations.
>
> We also conduct experiments by adding extra blocks for the domain-estimation branch $\theta_d$ to ensure that the training pipeline shares the same amount of parameters for different branching locations. Note we do not add parameters in the shared part and the target branch, since those parts are used at the test time. Adding more parameters to them will make an unfair comparison against the ERM baseline model. The results are listed in the following table. The observations are in line with the findings in our manuscript: the early-branching structure still performs better than the common practice with light-weight prediction heads when using the same amount of parameters. Please also refer to Section B.4 in our appendix for detailed descriptions.
>
> |&emsp;&emsp;&emsp;&emsp;&emsp;&emsp;&emsp;&emsp;&emsp;&emsp;|art&emsp;&emsp;&emsp;&emsp;&emsp;&emsp;|cartoon&emsp;&emsp;&emsp;&emsp;|photo&emsp;&emsp;&emsp;&emsp;|sketch&emsp;&emsp;&emsp;&emsp;|Avg.&emsp;&emsp;&emsp;&emsp;|
> |---------|--------|---|---|----|---|
> |ERM|78.0 $\pm$ 1.3|73.4 $\pm$ 0.8|94.1 $\pm$ 0.4|73.6 $\pm$ 2.2|79.8 $\pm$ 0.4|
> |$\theta_b=\textbf{I}$| 79.1 $\pm$ 0.7|74.1 $\pm$ 1.1 |94.6 $\pm$ 0.4|74.8 $\pm$ 0.6 |80.7 $\pm$ 0.5|
> |$\theta_b=bck_{1}$|80.5 $\pm$ 1.4| 74.1 $\pm$ 0.9|94.2 $\pm$ 0.5 |75.6 $\pm$ 0.3|81.1 $\pm$ 0.5|
> |$\theta_b=bck_{1,2}$|78.5 $\pm$ 1.1| 74.0 $\pm$ 0.7|94.4 $\pm$ 0.4 |75.7 $\pm$ 1.2|80.6 $\pm$ 0.4|
> |$\theta_b=bck_{1,2,3}$|79.2 $\pm$ 0.7|72.4 $\pm$ 2.1|93.7 $\pm$ 0.5|75.0 $\pm$ 1.2|80.1 $\pm$ 0.6|
> |$\theta_b=bck_{1,2,3,4}$|76.0 $\pm$ 0.9| 71.6 $\pm$ 0.7|93.5 $\pm$ 0.3 |71.6 $\pm$ 0.6|78.2 $\pm$ 0.5|
>
>
> * **2. Discussion of the results from the naive sampling and our RDS augmentation strategy**
>
> Compared to the naive sampling strategy without Eq. (7), the improvements brought by our RDS augmentation strategy are more noticeable in the "sketch" domain than in the rest "art", "cartoon", and "photo" domains. One possible reason is that images from the rest three domains are close according to the analysis in [1], and thus may not require too much diversity for augmentation when using one as the target domain. Differently, images from the "sketch" domain are distinct from others in their style statistics. When the "sketch" is the unseen domain, enabling Eq. (7) can select more diverse styles from the source domains compared to the naive model without it, leading to a more noticeable improvement accordingly. Following your suggestion, we include this discussion in our revised manuscript.
>
>
> * **3. Discuss a related work**
>
>
> Thanks for the notification. We discuss the mentioned art in our related works in the revised manuscript.
>
>
>
> * **4. Highlight the best and second best results in Table 2**
>
>
> Following your suggestion, we highlight the best and second results of different models in Table 2 for better viewing.
>
>
> $\textbf{Reference}$
>
> [1] Zhou, Kaiyang, et al. "Domain generalization with mixstyle". In ICLR, 2021.

---

> ### Author Response · Authors · 2022-11-22
> **Sincerely Look Forward to Your Feedback!**
>
> Dear Reviewer VzKA,
>
> Thanks again for your thoughtful suggestions to improve our work.
>
> We have carefully studied your comments and added additional experiments and analyses in our previous responses to address your concerns. We genuinely hope you could kindly check our responses.
>
> We hope that the new experiments and additional explanations have addressed your concerns. Please do not hesitate to contact us if there are other clarifications or experiments we can offer.
>
> Best wishes,
>
> Authors

---

> ### Author Response · Authors · 2022-12-06
> **Sincerely Look Forward to Your Feedback!**
>
> Dear Reviewer VzKA,
>
> We hope this finds you well. We have provided additional experiments and conceptual justifications to clarify your main concern regarding the experimental setting with the same amount of parameters in Sec. B.4 and Table 9 in the revised manuscript. Since the discussion stage is ending, we sincerely look forward to hearing back from you before the DDL.
>
> Best wishes,
>
> Authors

---

### Official Review · Reviewer_RNoa · 2022-10-24

**Confidence:** 5
**Correctness:** 3
**Technical Novelty And Significance:** 3
**Empirical Novelty And Significance:** 2
**Recommendation:** 3

**Clarity, Quality, Novelty And Reproducibility:**

Most of the parts are clear and well-written. Novelty is limited. Reproducibility is fine although there's no code, but one can easily reproduce it.

**Strength And Weaknesses:**

### Strength

1. Identifying the sharing and separate layers for dual-branch networks is interesting idea.
2. The paper is well-written.
3. The experiments are sufficient, with good performance.

### Weakness

1. The approach is not novel. Empirically finding the sharing layers can not be regarded as a technical contribution. For the second contribution (new augmentation), it is novel, but still very naive.
2. In page 4, authors spent too many contents explaining the initial results of identifying sharing layers, which I think is useless. It is just experimental observation. Plus, it might not be the case for other backbones like ResNet-50 and Vision transformer.
3. With regards to backbones, authors used DomainNet, which is good, but they only experiment on ResNet-18, which is rather outdated these days since ResNet-50 is the standard and more popular in DomainBed.
4. Identifying which layers to share is ad hoc. You should do everything again in face of a new backbone. This is the limitation of this work and cannot be called a contribution.
5. It will be benfeicial to see the actual running time of this approach compared to others.

**Summary Of The Paper:**

This paper proposes a domain dual branch network for domain generalization. The first contribution is to empirically verify which layers should be shared between domain and target classification. Then, authors propose a new augmentation strategy for domain augmentation. Experiments using ResNet-18 demonstrates its effectiveness.

**Summary Of The Review:**

This approach is on the right track: identifying the sharing and non-sharable layers between dual-branch networks. But the method is rather ad hoc and I don't think augmentation part is related to this paper. Plus, experimental results are not sufficient since only resnet18 is used.

---

> ### Author Response · Authors · 2022-11-16
> **Contributions of the early-branching observation and experiments with other backbones (1/2)**
>
> We thank the reviewer for the comments and we answer the raised questions below.
>
>
>
> * **1. Contribution of the early-branching observation**
>
>
> We are glad that the reviewer agrees that the idea of finding an appropriate branching location is interesting and on the right track. The design of using dual-branch networks to learn domain-invariant features is not rare in the literature [1, 2]. The common practice is to use a shared feature extractor with two light-weight prediction heads for the job. However, we observe that the common practice is detrimental to the performance and an early-branching structure, where the domain classification and target classification branches share the first few blocks while diverging thereafter, leads to better results. To the best of our knowledge, this observation is new and has not been reported in the literature. Also, as will be seen from our further experiments in the following response, we discover that the early-branching observation also applies to different network structures. Finally, we have provided some explanation in Page 4 to interpret this phenomenon. Based on the above reasons, we believe our early-branching observation is not trivial and can be inspiring for future work.
>
>
>
> * **2. Branching locations in other network structures**
>
>  Besides the observations on the ResNet18 backbone, we also conduct experiments using a much smaller network (i.e. a 4 layer network), the mentioned ResNet50 backbone, to show the performances from different branching locations. Following your suggestion, we further conduct experiments using vision transformers, including the DeiT [3] and T2T [4] models. Results listed in Table 4, 5, and 6 in the appendix are in line with our observation on page 4: the early-branching structure can lead to better performances. Please also refer to Sec. B.1 in the appendix for detailed descriptions. Inspired by these results, we suggest using an early-branching structure to obtain the domain-invariant features when given a dual-branch network in future research.
>
>
>
> * **3. Results on the ResNet50 backbone**
>
>
>
> Similar to the OoD-Bench [5], we mainly use the ResNet18 backbone as it can enlarge the gap in DG for different methods compared to a ResNet50 backbone. Because larger models are generally more robust to data from different domains, and thus the performance is easier to saturate on small datasets [6]. To further evaluate our method, we also conduct experiments on the ResNet50 backbone. We directly cite the results from existing works [7] for the compared arts and reevaluate ERM and our methods. Results are listed in Table 10 in the appendix. We report results from the top 3 arts and their ERM in the top 4 rows in the following table, and we report results from our ERM and our method in the bottom two rows. We observe that our method can surpass our ERM by 1.1 in average accuracy, while the top 3 arts lead their ERM by 1.3, 0.9, and 0.9, respectively. These results show that our method can obtain favorable performance against the strong baseline of ERM and existing arts when using the ResNet50 backbone.
>
>
> |&emsp;&emsp;&emsp;&emsp;&emsp;&emsp;&emsp;&emsp;&emsp;&emsp;|PACS&emsp;&emsp;&emsp;&emsp;|VLCS&emsp;&emsp;&emsp;&emsp;|OfficeHome|TerraInc&emsp;&emsp;|DomainNet|Avg.|
> |---------|--------|---|---|----|---|-----|
> |ERM (from [7] )|85.5 $\pm$ 0.2|77.5 $\pm$ 0.4|66.5 $\pm$ 0.3|46.1 $\pm$ 1.8|40.9 $\pm$ 0.1|63.3|
> |CORAL (from [7] )|86.2 $\pm$ 0.3| 78.8 $\pm$ 0.6|68.7 $\pm$ 0.3 |47.6 $\pm$ 1.0|41.5 $\pm$ 0.1 |64.6|
> |SagNet (from [7] )|86.3 $\pm$ 0.2| 77.8 $\pm$ 0.5|68.1 $\pm$ 0.1 |48.6 $\pm$ 1.0|40.3 $\pm$ 0.1 |64.2|
> |SelfReg (from [7] )|85.6 $\pm$ 0.4| 77.8 $\pm$ 0.9|67.9 $\pm$ 0.7 |47.0 $\pm$ 0.3|42.8 $\pm$ 0.0 |64.2|
> |ERM |83.1 $\pm$ 0.9|77.7 $\pm$ 0.8|65.8 $\pm$ 0.3|46.5 $\pm$ 0.9|40.8 $\pm$ 0.2|62.8|
> |Ours|85.9 $\pm$ 0.4| 78.8 $\pm$ 0.4|68.4 $\pm$ 0.3 |45.7 $\pm$ 1.4|40.5 $\pm$ 0.4 |63.9|
>
> * **4. Confronting a new backbone**
>
> We validate our observation using 5 commonly-used backbones, including both CNNs and ViTs, and all the results are in line with the observation that an early-branching structure is beneficial for the performance. It is highly possible that we may observe the same when confronting a new backbone. This phenomenon can be explained by the role of the base feature extractor $\theta_b$: if the two branches share a significant part of $\theta_b$. Then $\theta_b$ will be asked to provide both domain and target information, making it harder to disentangle them at the respective branches. However, if an early-branching architecture is used, we could minimize the need for encoding domain information in the pathway of target prediction. Thus the early-branching structure can be more effective in practice.

---

> > ### Author Response · Authors · 2022-11-16
> > **Contributions of the early-branching observation and experiments with other backbones (2/2)**
> >
> > * **5. Training time comparisons**
> >
> > Same as most DG methods, our model uses the same resource with ERM during the test phase. As listed in our limitation, since our method requires another thread for the domain-specific classification task during training, it will inevitably bring extra effort to the system. The average training time (TT) (minutes) of one trial in PACS for different methods with a same V100 GPU node are listed below. Following your suggestion, we also include the time comparisons in Table 13 in our appendix. Note that in DomainBed, some methods may use fewer updating steps for their main networks, smaller training batches, or fewer backward samples than the ERM method, thus requiring less training time than ERM.
> >
> > ||TT||TT||TT|
> > |---------|--------|---|---|----|---|
> > |DANN|17|GroupDRO|24|MixStyle|25|
> > |VREx|17|CORAL |24 |SD|25 |
> > |Fishr|17| CDANN|24 |ERM+HSIC|25|
> > |IRM|18|SagNet |24 |CondCAD| 26 |
> > |Mixup|18|CAD|24|$\textbf{Ours}$|29|
> > |MTL|18|ARM|24|MLDG|32|
> > |MMD|18|RSC|25|Fish|52 |
> > |$\textbf{ERM}$|24|SelfReg|25|
> >
> >
> > * **6. Reasons for including the augmentation strategy in our method**
> >
> > The augmentation strategy is a rational and effective component of our method for the following reasons.
> >
> > First, as stated in Section 2, using augmentation to improve the robustness is a common practice in DG. Following the prevalent design, we also incorporate an augmentation strategy in the overall framework, which is agreed to be novel and also effective as suggested in our analysis.
> >
> > Second, our work aims to disentangle domain-invariant and domain-specific features by specifically enforcing an independent constraint among them. However, the limited domain-specific information provided by the original data may easily overfit the target domain-invariant features. To enable the target features to be independent against various unknown domain-specific information during the test phase, we thus use augmentation to enlarge the domain information during training. This reason is also stated in Section 1, and 3.2 in our manuscript.
> >
> > Third, we want to stress that our augmentation method is not trivially packed into the overall framework, and it can fully exploit the benefit of the early-branching-based design. This is because we use augmented view to further construct the sensitivity loss in Eq. (3) and augmented independent constaint in Eq. (4). With augmentation and those two losses, we can further encourage the branches to encode different domain-invariant and domain-specific information. In this sense, the augmentation part is an important part of our method. Our ablation studies in the manuscript and appendix also indicate that these designs can further benefit the performance. We note using these designs to combine the augmentation strategy is also appreciated by Reviewer sawe.
> >
> >
> > $\textbf{Reference}$
> >
> > [1] Chen, Yang, et al. "A style and semantic memory mechanism for domain generalization". In ICCV, 2021.
> >
> > [2] Atzmon, Yuval, et al. "A causal view of compositional zero-shot recognition". In NeurIPS, 2020.
> >
> > [3] Touvron, Hugo, et al. "Training data-efficient image transformers \& distillation through attention". In ICML, 2021.
> >
> > [4] Yuan, Li, et al. "Tokens-to-token vit: Training vision transformers from scratch on imagenet". In ICCV, 2021.
> >
> > [5] Ye, Nanyang, et al. "OoD-Bench: Quantifying and Understanding Two Dimensions of Out-of-Distribution Generalization". In CVPR, 2022.
> >
> > [6] Hendrycks, Dan, et al. "The many faces of robustness: A critical analysis of out-of-distribution generalization". In ICCV, 2021.
> >
> > [7] Kim, Daehee, et al. "Selfreg: Self-supervised contrastive regularization for domain generalization". In ICCV, 2021

---

> ### Author Response · Authors · 2022-11-22
> **Sincerely Look Forward to Your Feedback!**
>
> Dear Reviewer RNoa,
>
> Thanks again for your thoughtful suggestions to improve our work.
>
> We have carefully studied your comments and added additional experiments and analyses in our previous responses to address your concerns. We genuinely hope you could kindly check our responses.
>
> We hope that the new experiments and additional explanations have convinced you of the merits of our work. Please do not hesitate to contact us if there are other clarifications or experiments we can offer.
>
> Best wishes,
>
> Authors

---

> ### Author Response · Authors · 2022-12-06
> **Sincerely Look Forward to Your Feedback!**
>
> Dear Reviewer RNoa,
>
> We hope this finds you well. Regarding your main concern about verifying the early-branching structures in different network structures, including ResNet50, and vision transformers, we hope the provided experiments in Tables 4, 5, and 6 have clarified your concern. Since the discussion stage is ending, we sincerely look forward to hearing back from you before the DDL.
>
> Best wishes,
>
> Authors

---

> > ### Comment · Reviewer_RNoa · 2022-12-11
> > **Further feedback**
> >
> > Reviewer would like to thank authors' efforts in providing a detailed response. However, the main issue is with technical novelty, which is not resolved by the response. Thus, I will keep my original rating.

---

### Official Review · Reviewer_FWNq · 2022-10-24

**Confidence:** 4
**Correctness:** 3
**Technical Novelty And Significance:** 2
**Empirical Novelty And Significance:** 2
**Recommendation:** 5

**Clarity, Quality, Novelty And Reproducibility:**

The paper is well-written and the method is well designed.
The quality of ablations could be substantially improved. The paper does demonstrate that each component of the proposed approach positively contributes to the results, however it doesn't effectively show that these components are superior to other reasonable alternatives.

Based on the paper I believe the work would be straightforward to implement and reproduce the results.

**Strength And Weaknesses:**

This paper presents a new approach for domain generalization through well reasoned design decisions and shows that the method outperforms the strong baseline of ERM and has the highest average accuracy amongst related methods.

The novel data augmentation strategy seems well reasoned and has strong potential, however it's unclear whether as a standalone it represents a useful contribution. Would be very beneficial to see alternate augmentation strategies combined with losses and branching strategies proposed in this work.

The results of the ERM+HSIC represents a weakness of this work, it is the core formulation of proposed approach yet shows marginal benefits over the baseline ERM.

A weakness with this work is their seems to be a conflation of hyperparameter search with the domain bed evaluation strategy, some parameters such as branching depth, alpha, and beta are chosen "empirically" in a manner that seems to invalidate the spirit of choosing hyperparameters over random trials in the validation pass over DomainBed.




**Summary Of The Paper:**

This work proposes a novel approach for domain generalization which relies on an early branching strategy to learn domain invariant features. They show that using HSIC as a domain invariant loss is most effective and also propose a new data augmentation strategy to simulate shifted domains. This work shows that this combination of design decision leads to state-of-the-art performance for domain generalization. While results clearly demonstrating the superiority of early branching for learning domain invariant representation would be quite consequential, the result presented in this work leave room for multiple interpretation as their is significant overlap in performance between branching at over blocks 0-3.

**Summary Of The Review:**

This is an interesting work with components that might be useful for future work in domain generalization, a very important problem.

However, presenting the results as combination of components makes its hard to distinguish if any individual component represents a meaningful and transferrable contribution. As the main goal is to improve the ability of machine learning model to generalize to unseen domains and not increase placement on leaderboards such as DomainBed its hard to determine the utility of this works contributions.

More detailed exploration of either Early Branching or RDS could greatly improve this work even if it doesn't yield a new state-of-the-art number.

---

> ### Author Response · Authors · 2022-11-16
> **Improved ablation studies and experiments with random hyper-parameters**
>
> We thank the reviewer for the comments and we answer the raised questions below. All discussions and results will be added to the revised manuscript.
>
>
>
> * **1. Improving ablation studies to evaluate the new RDS augmentation method**
>
> Following your suggestion, we revised our ablation studies by combining the evaluated part with other designs. Results in Table 3 show that our RDS strategy performs favorably against other arts.
>
>
>
> * **2. Performance of the model ERM+HSIC**
>
> First, we want to stress that (1) ERM is a strong baseline in DomainBed as discovered in [1], which is also agreed by the reviewer; (2) the final result is the average of multiple datasets. For some datasets, the differences between different methods are generally less significant. This will make the average accuracy close to each other for different methods. For reasons 1 and 2, we can see that existing methods usually achieve only marginal improvements against ERM, and more than half of them are inferior to the simple design of the model ERM+HSIC.
>
> Moreover, when compared with ERM, ERM+HSIC leads in 4 out of 5 datasets (i.e. PACS, VLCS, OfficeHome, and DomainNet) and ties in the 5th (i.e. TerraInc). These results are not trivial and can clearly demonstrate that the simple independent constraint can improve DG, which may be beneficial for future research.
>
> Last but not least, although the average improvement is less significant for the design, we show that it can obtain competitive performance combined with our RDS augmentation method. With a potentially more effective augmentation strategy, we believe the simple design can be further strengthened in future works.
>
>
>
> * **3. Experiments with random hyper-parameter settings**
>
>
>
> Following your suggestion, we reevaluate our methods with random settings of the hyper-parameters $\alpha$ and $\beta$ in Eq. (5), which are randomly selected from the range of [0.03, 0.3] for $\alpha$ and [0.1, 1] for $\beta$ according to Table 12 in the appendix.  All results (including Ours and ERM+HSIC) that involved these two hyper-parameters have been updated in our manuscript and appendix.
>
> We do not use random branching depth for our experiments because we find the early-branching structure is better than other options on almost all occasions according to our analysis in Section 3.1 in the manuscript and Section B in the appendix. We also conduct experiments to show the differences between using random branching depth and our early-branching structure in the following. Note we do not use the options with $\theta_{b}=\textbf{I}$ and $\theta_{b}=bck_{1,2,3,4}$ in the random branching depth experiment since our RDS is not compatible with the former setting and the latter can obviously deteriorate the performance according to our analysis.
>
>
>
> |&emsp;&emsp;&emsp;&emsp;&emsp;&emsp;&emsp;&emsp;&emsp;&emsp;|PACS&emsp;&emsp;&emsp;&emsp;|VLCS&emsp;&emsp;&emsp;&emsp;|OfficeHome|TerraInc&emsp;&emsp;|DomainNet|Avg.|
> |---------|--------|---|---|----|---|-----|
> |Random depth|82.0 $\pm$ 0.4|76.6 $\pm$ 0.3|61.7 $\pm$ 0.1|42.8 $\pm$ 1.4|35.1 $\pm$ 0.1|59.6|
> |Early-branching|82.4 $\pm$ 0.4| 76.5 $\pm$ 0.4|62.2 $\pm$ 0.1 |43.2 $\pm$ 1.3|34.9 $\pm$ 0.1 |59.8
>
> We observe that the performance from random depth is inferior to that from our early-branching structure in 3 out of 5 datasets. We thus use the early-branching structure in our experiment.
>
> * **4. More explorations of the early-branching structure and RDS augmentation strategy**
>
> Following your suggestion, we conduct more experiments to study the early-branching structure and RDS strategy in our appendix. Specifically, in Section B, we use varying network structures (including smaller and larger convolution networks and vision transformers), and different independent constraints (including orthogonal and correlation constraints), and combine them with our augmentation method to further evaluate the early-branching structure. Results are listed in Table 4-8. The common findings in these analyses suggest that the early-branching structure is superior to other options. In Section C.5, we visualize the augmented style statistics to better illustrate the differences between the compared augmentation strategies. The plotted features indicate that our augmented style statistics are more diverse than other alternatives. Please also refer to the text for details.
>
> $\textbf{Reference}$
>
> [1] Gulrajani, Ishaan and Lopez-Paz, David. "In search of lost domain generalization". In ICLR, 2021.

---

> ### Author Response · Authors · 2022-11-22
> **Sincerely Look Forward to Your Feedback!**
>
> Dear Reviewer FWNq,
>
> Thanks again for your thoughtful suggestions to improve our work.
>
> We have carefully studied your comments and added additional experiments and analyses in our previous responses to address your concerns. We genuinely hope you could kindly check our responses.
>
> We hope that the new experiments and additional explanations have addressed your concerns. Please do not hesitate to contact us if there are other clarifications or experiments we can offer.
>
> Best wishes,
>
> Authors

---

> ### Author Response · Authors · 2022-12-06
> **Sincerely Look Forward to Your Feedback!**
>
> Dear Reviewer FWNq,
>
> We hope that we have resolved the major concerns from your side. In particular, regarding the reformulation of the ablation study part, we hope the new formulation in Table 3 has clarified your concerns. Since the discussion stage is ending, we sincerely look forward to hearing back from you before the DDL.
>
> Best wishes,
>
> Authors

---

### Author Response · Authors · 2022-11-16
**General response**

We sincerely appreciate all reviewers' efforts in reviewing our paper and giving insightful comments and thoughtful suggestions toward improving our manuscript. We have revised our paper accordingly, and the major revisions can be summarized as follows.

* **Experiments with random hyper-parameters.** We reevaluate our method by randomly selecting hyper-parameters (from a certain range) in DomainBed and update the corresponding results in all tables from the manuscript and appendix.

 * **Improving ablation study.** We improve the ablation study by evaluating more combinations of the key parts of the proposed method.

* **More analysis and results.** We conduct more explorations regarding the early-branching structure by examining it with more network structures and different settings, and we also study our RDS augmentation strategy by visually comparing it with different arts. Details can be found in Sections B and C in the appendix. The newly added experiments further validate our claim and answer the reviewer's questions.

Our detailed responses can be found in the following.

---

### Decision · Program_Chairs · 2023-01-20

**Decision:**

Reject

**Justification For Why Not Higher Score:**

See above.

**Justification For Why Not Lower Score:**

N/A

**Metareview: Summary, Strengths And Weaknesses:**

We had a video call with reviewers to discuss this paper and reach a conclusion that this paper still needs further improvement. First, the observation that we need an early branching strategy to learn domain invariant features is heuristic. It would be great if authors can provide more principle guidelines to choose which layer to branch out the network and how this is sensitive to different network architectures with different depth. Second, the proposed augmentation technique is not well motivated by the major observation introduced in the beginning. Instead, they are more like two independent components which can empirically improve the performance.

**Summary Of Ac-Reviewer Meeting:**

All reviewers agree that the technical contributions of this work is close to borderline. Considering the major finding in the work is empirical and heuristic without fundamental principles, and the two components of the final method are not well connected to each other, we agree this is a borderline leaning towards reject.